# Signatures of $V_H1$-69-derived hepatitis C virus neutralizing antibody precursors defined by binding to envelope glycoproteins

Joan Capella-Pujol [1,2], Marlon de Gast[1,2], Laura Radić [1,2], Ian Zon [1,2], Ana Chumbe [1,2], Sylvie Koekkoek[1,2], Wouter Olijhoek[1,2], Janke Schinkel[1,2], Marit J. van Gils [1,2], Rogier W. Sanders [1,2,3] ✉ & Kwinten Sliepen [1,2] ✉

An effective preventive vaccine for hepatitis C virus (HCV) remains a major unmet need. Antigenic region 3 (AR3) on the E1E2 envelope glycoprotein complex overlaps with the CD81 receptor binding site and represents an important epitope for broadly neutralizing antibodies (bNAbs) and is therefore important for HCV vaccine design. Most AR3 bNAbs utilize the $V_H1$-69 gene and share structural features that define the AR3C-class of HCV bNAbs. In this work, we identify recombinant HCV glycoproteins based on a permuted E2E1 trimer design that bind to the inferred $V_H1$-69 germline precursors of AR3C-class bNAbs. When presented on nanoparticles, these recombinant E2E1 glycoproteins efficiently activate B cells expressing inferred germline AR3C-class bNAb precursors as B cell receptors. Furthermore, we identify critical signatures in three AR3C-class bNAbs that represent two subclasses of AR3C-class bNAbs that will allow refined protein design. These results provide a framework for germline-targeting vaccine design strategies against HCV.

Hepatitis C virus (HCV) continues to be a major public health problem, with about 58 million people living with chronic HCV infection. The World Health Organization (WHO) estimates that almost 300,000 people die annually from HCV-related causes such as cirrhosis and hepatocellular carcinoma[1]. While direct-acting antivirals (DAAs) may cure more than 95% of people with HCV, several factors, such as the high cost of antivirals, viral resistance, the risk of re-infection, and the lack of access to diagnosis hamper global HCV eradication programs[2]. Therefore, an effective preventive vaccine for HCV is a major unmet need[3,4].

Such a preventive HCV vaccine should probably elicit neutralizing antibodies (NAbs), since induction of (cross-reactive) NAbs in infected individuals correlates with protection against re-infection and viral clearance[5-8]. The extreme genetic variability of the virus is a major barrier, particularly in the envelope glycoprotein E1E2, the only target for NAbs[9]. The heavy glycosylation, conformational flexibility and heterogeneity of E1E2 provide further obstacles for vaccine development[10,11]. Nevertheless, broadly neutralizing antibodies (bNAbs) have been isolated from HCV-infected individuals and are able to surmount these challenges by targeting conserved epitopes on E1 or E2[12,13]. These bNAbs not only provide protection against infection, but can also clear an established infection[5-7,14-18].

E2 is the receptor binding-subunit and interacts with scavenger-receptor class 1 B (SR-B1) and tetraspanin CD81 on the host cell to mediate cell entry[19-21]. The neutralizing face of E2 is the most common target for bNAbs, and it consists of the CD81 binding site (CD81bs) and several other antigenic clusters[22,23]. Most bNAbs that target these regions impede interaction of the virus with the host cell receptor CD81, particularly Abs targeting antigenic region (AR) 3, which overlaps with the CD81bs[13]. Cross-genotype bNAbs found in patients predominantly target AR3, often requiring little somatic hypermutation (SHM) to acquire breadth, making this region an appealing target for vaccine design[7,14].

[1]Department of Medical Microbiology and Infection Prevention, Laboratory of Experimental Virology, Amsterdam UMC, University of Amsterdam, 1105 AZ Amsterdam, Netherlands. [2]Amsterdam Institute for Infection and Immunity, Infectious Diseases, 1105 AZ Amsterdam, Netherlands. [3]Department of Microbiology and Immunology, Weill Medical College of Cornell University, New York, NY 10065, USA. ✉e-mail: r.w.sanders@amsterdamumc.nl; k.h.sliepen@amsterdamumc.nl

The vaccine approach known as "germline-targeting" is an attractive strategy to overcome HCV's diversity and induce bNAbs. Germline-targeting approaches have been pioneered in the HIV-1 field and three germline-targeting HIV-1 immunogens are currently being evaluated in phase 1 clinical trials (clinicaltrial.gov: NCT03547245, NCT04224701, NCT05471076)[24–26]. These strategies try to reproduce the natural affinity maturation of B cells from naive germline state to mature B cells. In order to elicit bNAbs, a germline-targeting vaccine strategy aims to prime specific bNAb-precursor B cells and boost them to induce the affinity-enhancing mutations desired to become mature bNAb B cells using a series of rationally designed immunogens. It is imperative then, that germline-reverted forms of bNAbs exhibit appreciable affinity for the priming immunogen(s). The major binding determinant of most bNAbs is their heavy chain (HC) complementarity-determining region 3 (CDRH3), which is encoded by different recombination of V-D-J genes[27]. Further variation is gained in the CDRH3 region as a product of the junctions between genes. Such exceptional diversity in the human B cell repertoire suggests that targeting a specific single bNAb-precursor is fairly impractical as a vaccine target. Instead, a pool of precursors sharing a set of bNAb-associated genetic features should be identified and targeted. Therefore, a germline-targeting immunogen should engage a number of bNAb precursors in order to induce the desired response in diverse vaccine recipients[28]. The first HIV-1 germline-targeting vaccines focused on the VRC01-class of bNAbs that target the CD4 binding site on the HIV-1 envelope glycoprotein. This class of bNAbs has been identified in several different HIV-1 infected individuals and is typified by the $V_H1$-2 origin (i.e., the V gene used by the B cells to generate the HC of their B cell receptor or BCR) of their HC coupled with a short 5 amino acid CDRL3 loop in the light chain (LC)[29–31].

The HC of many AR3-directed antibodies (Abs) share genetic properties as most of them are derived from the $V_H1$-69 gene family[32]. AR3C is one of the most potent members of this class of $V_H1$-69-derived bNAbs and one of the first members to be identified of what we refer to as the AR3C-class of bNAbs. Other members include, but are not limited to, AR3A, AR3B, AR3D, HEPC3, HEPC74, and AT1209[14,17,33]. The involvement in viral immunity of antibody precursors of this gene family goes beyond HCV as it has also been associated with cross-neutralization of HIV-1[34,35], influenza virus[36–38], dengue virus[39], and SARS-CoV-2[40–42]. Thus, the commonality of the $V_H1$-69 precursor gene, the breadth and potency of several of these bNAbs and the ability of different patients to raise these bNAbs, make the $V_H1$-69-derived bNAbs desirable targets for germline-targeting vaccines. However, $V_H1$-69 germline-targeting vaccine strategies for HCV have been hampered by the poor understanding of the E1E2 glycoprotein function and structure, the insufficient number of E1E2 protein reagents that bind $V_H1$-69-derived germline precursors, and the lack of well-defined shared sequence signatures within and outside the CDRH3.

Here, we identify E2E1 trimer and nanoparticle immunogens that bind to inferred AR3C-class antibody precursors and are capable of activating B cells expressing such bNAb precursors as their BCR. Furthermore, we established that the binding of inferred germline bNAbs relies heavily on the CDRH3, as shown for gl-AR3C by retention of binding after LC exchange and CDRH3 engraftment into different $V_H$ backgrounds. Finally, we found that the allelic $V_H1$-69 environment is critical for one subclass of AR3C-class precursors but not another. This study therefore provides a framework for the rational design of an HCV germline-targeting immunogen focused on inducing AR3C-class bNAbs.

## Results

### Inferred germline AR3C-class antibodies bind to selected HCV E2E1 trimers

A germline-targeting vaccination approach requires an antigen that interacts with the desired germline bNAbs in order to study their binding requirements. However, most inferred germline AR3C-class bNAbs usually do not bind recombinant nor membrane-bound HCV glycoproteins and/or show no or very weak neutralizing capacity[14,32,43]. We set out to identify E1E2 antigens capable of binding AR3C-class bNAb precursors efficiently.

First, we generated a panel of inferred germline precursors (gl) of AR3C-class bNAbs: AR3A, AR3B, AR3C, AR3D, HEPC3, HEPC74, and AT1209. Most inferred germline sequences were published previously[32,43], while the others were computationally inferred using IMGT/V-QUEST software, a tool that predicts the most likely VDJ rearrangement based on the DNA sequence of an antibody[44]. We generated two versions of gl-AT1209: one with and one without a duplicated sequence found in the CDRH2 (see below, Supplementary Fig. 1a)[23,33].

Next, we tested binding of these inferred germline AR3C-class precursors and their mature counterparts to a panel of recombinant HCV glycoprotein trimers derived from seven different HCV strains covering 6 genotypes (genotype 1a: H77; genotype 2a: UKNP2.2.1; genotype 2b: AMS2b; genotype 3a: AMS3a; genotype 4a: UKNP4.1.1; genotype 5: UKNP5.1.2; and genotype 6: UKNP6.2.1). To generate these HCV glycoprotein trimers, a novel E1E2 permutation design was applied in which the gene segments encoding the E2 and E1 ectodomains were reversed and the subunits were separated using a furin cleavage site[45]. The E2E1 heterodimers were genetically fused to an I53-50A trimerization domain to generate E2E1 trimers (Fig. 1a, see "Methods"). The mature bNAbs bound efficiently to all seven E2E1 trimers in ELISA (Fig. 1b, c and Supplementary Fig. 1b). AT1209, AR3C, and HEPC74 showed significantly stronger binding than HEPC3 and AR3D (Supplementary Fig. 1b, Friedman test, $p < 0.05$). Overall, mature bNAbs bound significantly stronger to E2E1 trimers derived from UKNP2.2.1 (genotype 2a) and UKNP4.1.1 (genotype 4a) compared to those from AMS2b, UKNP5.2.1, and UKNP6.1.2 (Friedman test, $p < 0.05$). As expected, germline bNAbs showed significantly weaker binding than their corresponding mature bNAbs ($p < 0.0001$, paired Wilcoxon signed-rank test, Fig. 1b). Six out of seven E2E1 trimers engaged gl-AR3C and five out of seven E2E1 trimers engaged gl-HEPC74 (Fig. 1b, Supplementary Fig.1c). gl-AR3A only engaged the UKNP5.2.1 E2E1 trimer, but the UKNP5.2.1 E2E1 trimer showed only very weak binding to gl-HEPC74 (Fig. 1c), which was confirmed by biolayer interferometry (BLI) measurements (Fig. 1d). Conversely, UKNP4.1.1 E2E1 showed relatively strong binding to gl-AR3C and gl-HEPC74 in ELISA and BLI (Supplementary Fig. 1c and Fig. 1c, d). We did not detect binding or very minimal binding of gl-AR3B, gl-AR3D, gl-HEPC3, and gl-AT1209 (with or without CDRH2 insert) to the trimers in our panel.

### Inferred germline AR3C-class antibodies neutralize selected HCV pseudoparticles

Next, we tested whether the germline precursors of AR3A, AR3C, and HEPC74, i.e., the best binders identified above, were also able to neutralize HCV. Therefore, we tested their neutralization capacity as well as their mature counterparts against a panel of HCV pseudoparticles (HCVpp) that were sequence-matched to our E2E1 trimers. As anticipated, the mature bNAbs neutralized all tested HCVpp strains and mature AR3C displayed the most potent neutralization (50% inhibitory dose ($IC_{50}$) ranging from 0.01 to 0.2 µg/mL, Fig. 1e, Supplementary Fig. 2a).

The three inferred germline antibodies neutralized some of the HCVpps tested and were overall less potent than the mature bNAbs ($p < 0.0001$, paired Wilcoxon signed-rank test, Fig. 1e). gl-AR3C neutralized all HCVpp strains tested with $IC_{50}$ values ranging from 0.2 to 35 µg/mL (Supplementary Fig. 2b). gl-HEPC74 neutralized five out of the seven HCVpp's tested (Supplementary Fig. 2b). gl-AR3A neutralized H77, while gl-HEPC74 did not. Conversely, gl-HEPC74 neutralized UKNP2.2.1 and UKNP4.1.1, which resisted neutralization by gl-AR3A. AMS3a HCVpp was neutralized by the three germline bNAbs ($IC_{50}$ 1.0–10 µg/mL), while AMS2b was only neutralized by gl-AR3C.

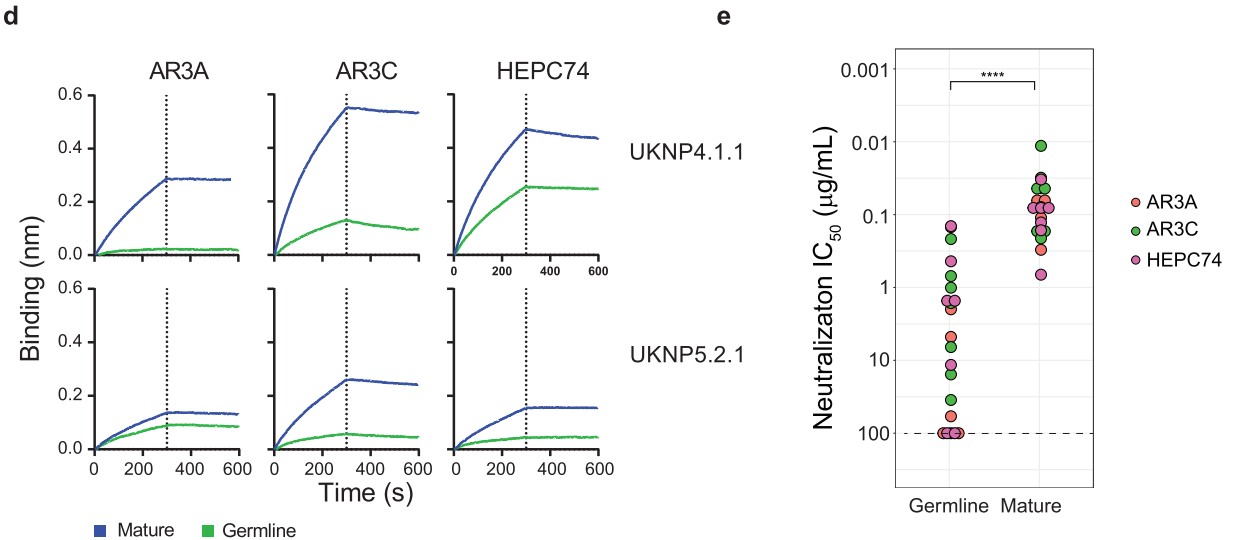

Overall, the neutralization data ($IC_{50}$) correlated well with the binding capacity of the antibodies (Supplementary Fig. 2c, Spearman's rank correlation $p < 0.0001$). Thus, germline antibodies with high binding activity, such as gl-AR3C and gl-HEPC74, neutralized HCVpp more potently (Supplementary Fig. 2). There were some exceptions: in some cases, germline bNAbs displayed neutralizing activity in the absence of detectable binding to the soluble trimer (e.g., gl-AR3A with AMS3a), pointing to subtle differences in binding and neutralization (Supplementary Fig. 2d). These discrepancies could originate from epitope differences resulting from the E2E1 trimer design or from using different cell lines for the production of E2E1 trimer and the production of HCVpp[46]. Our data suggests that HCV immunogens based on selected E1E2 sequences, such as those from isolates UKNP4.1.1 and UKNP5.2.1, provide lead candidates that might not require (many) additional mutations to function as an AR3C-class germline-targeting immunogen.

**Fig. 1 | Binding and neutralization by germline AR3C-class bNAbs. a** Top: linear representation of full-length E1E2 with the hypervariable region 1 (HVR1), transmembrane domains (TMD) and hydrophobic external region (MPER) indicated. Bottom: schematic representation of the permuted E2E1 trimer design. E1 and I53-50A.NT1 (I53-50A) are separated by a Gly-Ser linker (GSGGSGGSGGSGGS). Numbering is based on standard H77 polyprotein numbering[100]. E2 and E1 are separated by a Gly-Ser linker (GGSGGSGGSGGSGGS) followed by a furin cleavage site (RRRRRR). The protein contains a strepII-tag (WSHPQFEK) used for affinity purification. **b** Binding of AR3C-class bNAbs and their inferred germline predecessors against seven different E2E1 trimers from HCV genotypes 1-6. ELISA results are represented as area under the curve (AUC). A no-binding threshold of 0.001 is set by the dashed line. A paired two-sided Wilcoxon signed-rank test was used for the comparison (****$P = 3.5e-15$). **c** Representative ELISA binding curves for mature and inferred germline versions of AR3A, AR3C, and HEPC74 against H77, UKNP4.1.1, and UKNP5.2.1 E2E1 trimers. All experiments were performed in duplo (indicated by error bars). **d** BLI analysis of antibody binding to UKNP4.1.1 and UKNP5.2.1 E2E1-I53-50A. The indicated bNAbs were immobilized onto protein A biosensors and incubated with 500 nM of E2E1 trimers. The representative binding curves from two independent experiments. **e** Potency (in midpoint neutralization concentrations (IC$_{50}$)) of germline and mature AR3A, AR3C, and HEPC74 against HCVpp clones derived from seven different strains (genotypes 1-6). A paired two-sided Wilcoxon signed-rank test was used for the comparison (****$P = 9.5e-07$).

## B cells expressing inferred germline AR3C and HEPC74 can be activated by E1E2

To evaluate whether selected E2E1 trimers can trigger activation of B cells bearing gl-AR3C and gl-HEPC74, we generated HCV-specific Ramos B cells that stably express gl-AR3C, gl-HEPC74 or their mature counterpart as their BCR, and measured their activation by E2E1 trimers by monitoring Ca$^{2+}$ influx in vitro (Supplementary Fig. 3a, b). Antigens based on the UKNP4.1.1 strain, which strongly bound to both gl-AR3C and gl-HEPC74 (Fig. 1b, Supplementary Fig. 1c), and the H77 strain, which did not bind to gl-HEPC74, were used for the analysis. The presence of the I53-50A domain on the E2E1 domain allowed for the assembly into icosahedral nanoparticles displaying twenty E2E1 trimers (E2E1-NP). These two-component E2E1-NPs were assembled by mixing the E2E1 trimers with a second pentameric component (I53-50B) in vitro[45,47–52]. Multivalent display of antigens has repeatedly been shown to enhance cognate B cell activation over soluble antigen by allowing for high avidity interactions with BCRs. Multimeric display has also been particularly useful in the context of a VRC01-class germline-targeting HIV-1 vaccine candidate[53].

UKNP4.1.1 E2E1-NPs activated gl-AR3C B cells (Fig. 2a) at 100 μg/ml, while both E2E1 trimers and H77 E2E1-NPs did not, despite the fact that soluble H77 and UKNP4.1.1 E2E1 trimers bound to the gl-AR3C bearing B cells (Supplementary Fig. 3c). The mature AR3C-expressing B cells already showed activation at 10 μg/ml of UKNP4.1.1 E2E1-NPs and were also activated by adding UKNP4.1.1 E2E1 trimers and H77 E2E1-NPs. Similarly, gl-HEPC74-expressing B cells were only activated by adding 100 μg/ml of UNKP 4.1.1 E2E1-NPs, but not by adding trimers or H77 E2E1-NPs. Mature HEPC74 B cells were efficiently activated by H77 and UKNP4.1.1 E2E1-NPs (Fig. 2b, Supplementary Fig. 3c).

These results show that AR3C-class precursor B cells can be targeted and activated by selected permutated E2E1 immunogens, especially those based on the UKNP4.1.1 strain. Finally, the data reinforce that multivalent display benefits the activation of cognate B cells, and the I53-50NP platform is suitable for presenting E2E1 proteins.

## AR3C-class bNAbs bind the E1E2 heterodimer through an extended CDRH3

To gain molecular insight into the binding of AR3C-class bNAbs to E1E2, we superimposed the available structures of these mAbs in the context of the recently solved E1E2 heterodimer[23] (Fig. 3a). The E1E2 structure was obtained in complex with AR3C-class bNAb AT1209, in addition to bNAbs AR4A and IGH505[12,13,23,33]. The epitopes of the AR3C-class bNAbs show considerable overlap, but the angles of approach differ substantially: up to 50° when comparing AR3D and HEPC74 (Fig. 3a, right panel). The $V_H1$-69-derived AR3C-class bNAbs antibodies have adopted a similar mode of recognition of the region that directly interacts with the CD81 receptor (residues 427, 442, and 529)[32,54] (Fig. 3a), employing a common interaction of the rather long CDRH3 loops (Fig. 3b, right panel), often characterized by the presence of a stabilizing disulfide bond within the loop (Supplementary Fig. 1a). Two subclasses of bNAbs can be identified based on the structure of the CDRH3 loop and the angle of approach (Fig. 3a, right panel): those with a bent CDRH3 (AR3A,

AR3B, AR3C, AR3D and AT1209) and those with a straight CDRH3 (HEPC3 and HEPC74).

The footprints for the CDRH1 and CDRH2 loops differ for the bNAbs with a straight CDRH3 compared to the bNAbs with a bent CDRH3[43] (Fig. 3a, right panel). The CDRH1 and CDRH2 loops of HEPC3 and HEPC74 contact the N-terminus of the E2 α1 helix and the portion of the E2 front layer between the α helix and variable region 2 (residues 446–448), whereas the CDRH1 and CDRH2 loops of AR3A, AR3B, AR3C, and AR3D recognize hydrophobic residues on the C terminus of the α1 helix (L427, F442, and W529) (Fig. 3a, left panel)[32]. AR3C-class antibodies derive from the same or similar allelic variants of $V_H1$-69 (Supplementary Table 1)[14,32], therefore the differences in the CDRH1 and CDRH2 are the result of SHM. $V_H1$-69 encodes two hydrophobic residues at the tip of the CDRH2 loop (typically, a canonical hydrophobic amino acid at position 53 and a critical phenylalanine at position 54) that provide a structural basis for recognizing conserved hydrophobic epitopes on viral envelope glycoproteins beyond HCV[55]. Some of the AR3C-class bNAbs acquired additional hydrophobic amino acids in the CDRH2 through SHM (G55A in AT1209, T56V in AR3B and G50A and Y59L in AR3D) (Supplementary Fig. 1a). In particular, mature bNAb AT1209 contains an unusual elongated CDRH2 sequence of 32 amino acids (Fig. 3b, left panel, Supplementary Fig. 1a), while the average human CDRH2 length is 17 amino acids (AA), and rarely longer than 20 amino acids (Fig. 3b, left panel)[56]. To our knowledge, AT1209 possesses the longest human CDRH2 described to date. This unusually long CDRH2 contributes significantly to the binding of AT1209 to E2[23]. We hypothesize that this CDRH2 insertion was generated by a duplication event involving the FWR2 sequence preceding the CDRH2, as the suspected duplicated CDRH2 segment of 45 nucleotides (15 amino acids) shows 77% sequence identity with $V_H1$-69*01 FWR2. This would represent a duplication of almost the entire FWR2 (Supplementary Fig. 1a). Insertions and deletions (indels) represent an additional B cell maturation pathway, complementary to SHM, especially in antibodies involved in the recognition of pathogens with high antigenic variability, such as HIV-1 and influenza virus, but insertions of more than 33 nucleotides are very rare[57–61].

## The heavy chain is sufficient for HCV engagement and neutralization by germline HEPC74 and AR3C

The binding of gl-AR3C and gl-HEPC74, representing the subclasses with either a bent or a straight CDRH3 loop, respectively, to the E2E1 proteins of AMS3a, UKNP2.2.1, UKNP4.1.1 UKNP.6.1.2, and H77 (the latter by gl-AR3C only), allowed us to define additional binding requirements than hitherto possible. First, to assess the importance of the HC of AR3C and HEPC74 for binding, we generated chimeric antibodies by co-expressing the HCs of germline or mature AR3C or HEPC74 with a heterologous LC, i.e., the one from HIV-1 bNAb VRC01. Likewise, we co-expressed the HC of VRC01 with the LCs of (gl-)AR3C and (gl-)HEPC74 to determine the importance of the LCs. The (gl-)AR3C HC – VRC01 LC chimeric antibodies produced efficiently (~10−20 mg/L yield) and efficient HC/LC pairing was confirmed by SDS (Supplementary Fig. 4a). However, production of (gl-)HEPC74 HC – VRC01 LC chimeras was less efficient (yield ~1−4 mg/L), which could be

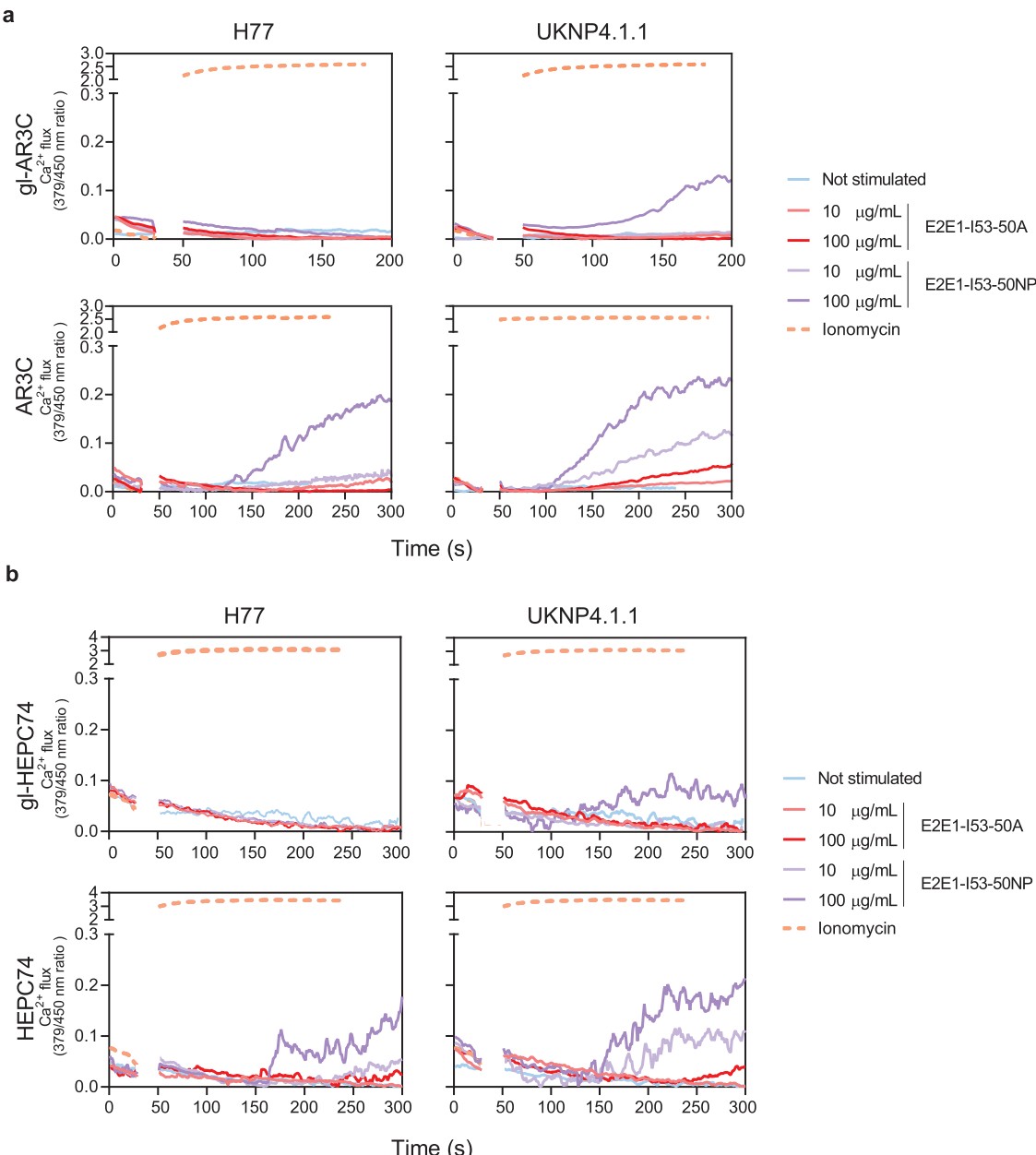

**Fig. 2 | Activation of B cells expressing mature and germline AR3C and HEPC74 by E2E1-I53-50A trimers and E2E1-I53-50NPs.** B cells expressing inferred germline (top) or mature antibodies (bottom) of AR3C (**a**) or HEPC74 (**b**) as BCRs were stimulated with either H77 or UKNP4.1.1 E2E1-I53-50A trimers (red), E2E1-I53-50 nanoparticles (purple), ionomycin (positive control, orange) or no stimulus (blue). The experiments were performed with 10 or 100 μg/mL E2E1-I53-50A or equimolar amount of E2E1-I53-50A in nanoparticles.

explained by the poor pairing efficiency observed in SDS gels (Supplementary Fig. 4b).

Chimeric AR3C HC – VRC01 LC engaged the above-described E2E1 trimers as efficiently as conventional AR3C (Fig. 4a). Similarly, no significant difference in binding was observed for the gl-AR3C HC – VRC01 LC chimeras compared to gl-AR3C. In contrast, chimeric HEPC74 HC – VRC01 LC and gl-HEPC74 HC- VRC01 LC showed significantly reduced, but detectable binding to E2E1 trimers compared to conventional HEPC74 and gl-HEPC74 (Fig. 4b). The loss of binding might be linked to the poor pairing efficiency of the (gl-) HEPC74 HC with a heterologous LC. As expected, chimeric (gl-)AR3C and (gl-) HEPC74 LC paired with VRC01 HC did not display any detectable binding.

Next, we performed HCVpp neutralization assays with the same set of chimeric antibodies. Chimeric AR3C HC – VRC01 – LC retained full neutralization capacity compared to conventional AR3C mirroring the results from our ELISA binding assays (Fig. 4a). Similarly, gl-AR3C – VRC01 LC showed no significant difference in neutralization capacity compared to conventional gl-AR3C (Fig. 4a).

Chimeric HEPC74 HC – VRC01 LC only neutralized the neutralization sensitive UKNP5.2.1 HCVpp at a >100-fold lower $IC_{50}$ value (4.4 μg/ml versus 0.03 μg/ml), while chimeric gl-HEPC74 HC – VRC01 LC did not neutralize any of the HCVpp's, mirroring the results of the binding ELISA (Fig. 4b).

Together, these results confirm that AR3C-class bNAbs rely heavily on the HC for binding and neutralization and that the LC only plays a minor role. This is consistent with observations that AR3C-class bNAbs utilize $V_H1$-69, but employ different LC genes[35,36,62–64].

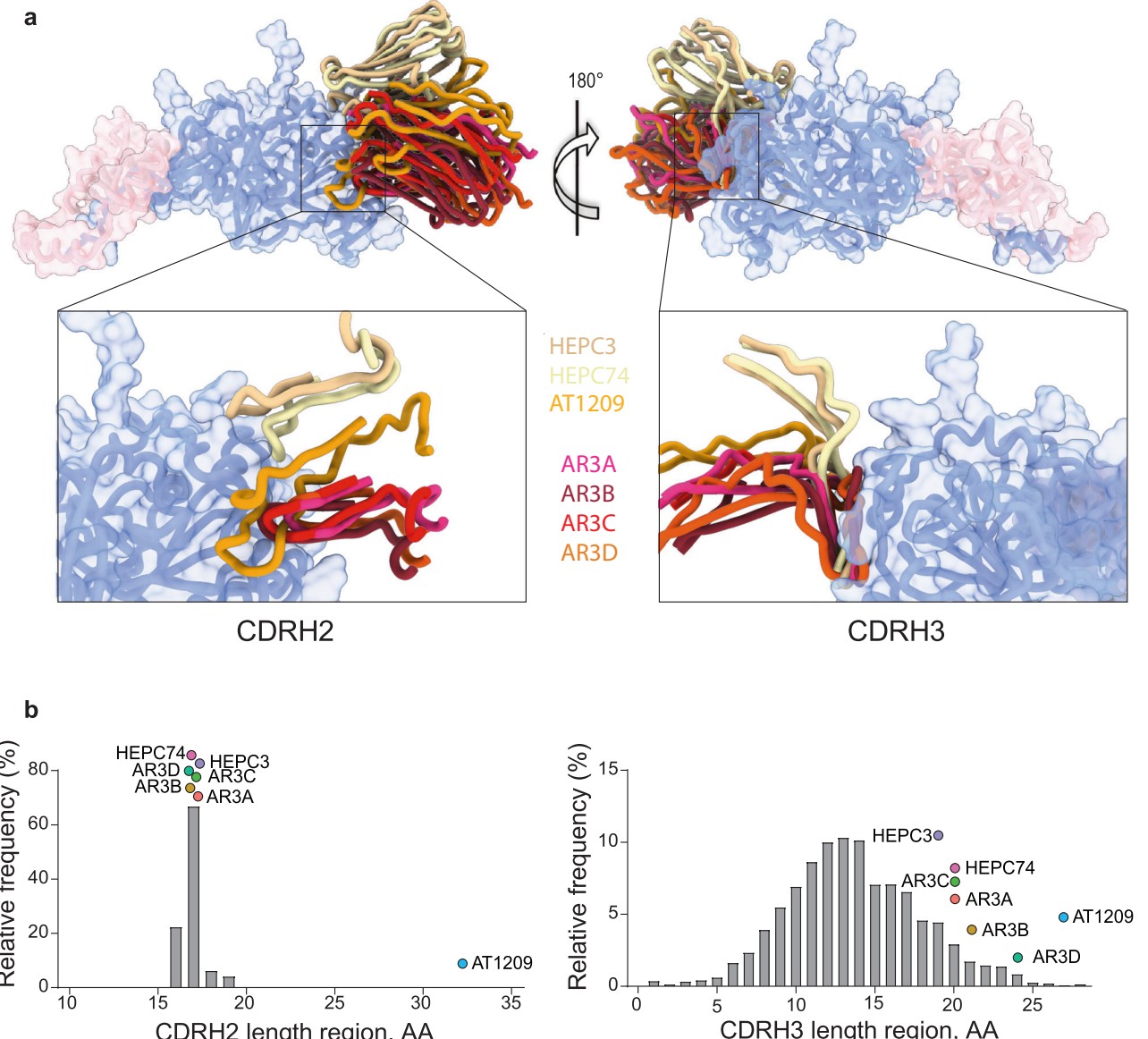

**Fig. 3 | Shared characteristics of AR3C-class bNAbs. a** Superposition of structures of AR3C-class bNAbs AR3C, AR3A, AR3B, AR3D, HEPC3, HEPC74, and AT1209 (PDB entries: 4MWF, 6BKB, 6BKC 6BKD 6MEI, 6MEH, 7T6X, respectively) on the E1E2 complex (PDB: 7T6X) which bind to the same AR3 epitope on the neutralizing face of E2. Only the variable heavy (VH) regions of the bNAbs are shown for clarity. Left panel: superposition of the CDRH2 (Kabat: 50-65). The exceptionally long CDRH2 of AT1209 is highlighted in orange. Right panel: superposition of the CDRH3 loops (Kabat: 91-104) of mature AR3C-class bNAbs with a bent (AR3A, AR3B, AR3C, AR3D, and AT1209) and straight CDRH3 (HEPC3 and HEPC74). **b** Length distribution of human CDRH2 (left) and CDRH3 (right). Human CDRH2 lengths were extracted from the online abYsis system (http://www.bioinf.org.uk/abysis/) using the Kabat numbering scheme Kabat and National Institutes of Health (U.S.). Highlighted are the relative positions of the AR3C-class antibodies.

## The $V_H1-69$ background is critical for binding by germline HEPC74 but not germline AR3C

While the structural analyses showed that the CDRH3 was responsible for the majority of the contacts, we wondered to what extent other factors encoded by $V_H1-69$ contributed to the binding. We therefore engrafted the germline AR3C and HEPC74 CDRH3-loops onto two $V_H$ genes that are dissimilar in sequence and CDRH2 hydrophobicity: $V_H1-2^*02$ and $V_H3-23^*01$ (Fig. 5a). $V_H1-2$ is a relatively common precursor gene and is used by HIV-1 bNAb VRC01[65], while $V_H3-23$ is one of the most commonly found $V_H$ genes in the naive human B cell repertoire and is used by several SARS-CoV-2 NAbs[40]. The gl-AR3C CDRH3 engrafted in $V_H1-2^*02$ and $V_H3-23^*01$ and gl-HEPC74 CDRH3 in $V_H1-2^*02$ yielded productive IgG (Supplementary Fig. 5a), but engrafting gl-HEPC74 CDRH3 into the $V_H3-23$ background did not (Supplementary Fig. 5b). The antibodies

that produced efficiently were then tested in ELISA for binding to the different E2E1 trimers.

Despite the low sequence similarity of $V_H1-69$ to $V_H1-2^*02$ or $V_H3-23^*01$ (79% and 57%, respectively), the gl-AR3C CDRH3 engrafted on $V_H1-2^*02$ or $V_H3-23^*02$ retained binding capacity to all E2E1 trimers, albeit binding was not as strong with as with conventional gl-AR3C (Fig. 5a). In contrast, the gl-HEPC74 CDRH3 lost binding when engrafted onto $V_H1-2^*02$ (Fig. 5a). Thus, gl-AR3C and gl-HEP74 have different requirements for the $V_H$ background and may or may not retain binding capacity depending on the environment.

## $V_H1-69$ allelic variation affects germline AR3C but not germline HEPC74

$V_H1-69$ is one of the most polymorphic loci within the human $V_H$ gene cluster, exhibiting both allelic and copy number variation[66,67]. Out of

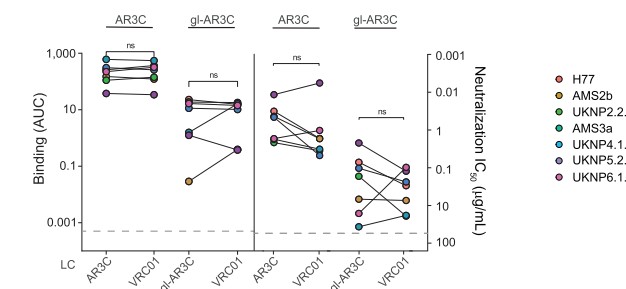

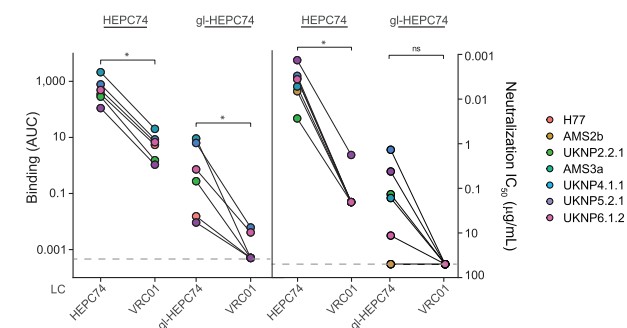

**Fig. 4 | Impact of the heavy and light chains on AR3C-class antibody binding and neutralization.** To determine the importance of the heavy chain (HC) and light chain (LC) of AR3C-class bNAbs for binding and neutralization we generated chimeric mAbs by pairing AR3C-class HCs with the LC of the heterologous anti-HIV-1 VRC01 mAb. **a** Binding and neutralization by chimeric (gl-)AR3C HC – VRC01 LC mAbs in ELISA (left) and HCVpp neutralization assays (right). **b** Similar as (**a**), ELISA binding (left) and HCVpp neutralization (right) by chimeric (gl-)HEPC74 HC – VRC01 LC to a panel of HCV strains. Binding signal is expressed as area under the curve (AUC) and neutralization activity is expressed as IC$_{50}$ (μg/mL). Maximum concentration tested for each chimeric antibody was 50 μg/mL. Paired two-sided Wilcoxon signed-rank test were used for all calculations (*$P < 0.05$, ns=non-significant). ELISA experiments where performed in duplo while neutralization assays were performed in triplo.

the 20 known $V_H1$-$69$ alleles, 13 contain a phenylalanine (F) at position 54 (Kabat numbering) and seven contain a leucine (L). Most $V_H1$-$69$ bNAbs (HCV and non-HCV), including AR3C-class antibodies, are derived from 'F alleles' containing F54[55]. Around 11% of individuals are homozygous for L alleles (L/L) containing L54[37], which implies that 11% of individuals might not be able to generate such bNAbs. To investigate the importance of the allelic variation of $V_H1$-$69$ for engagement of germline AR3C ($V_H1$-$69$*06 derived) and HEPC74 ($V_H1$-$69$*01 derived), we engrafted their CDRH3 sequences onto six of the most frequent $V_H1$-$69$ alleles including both F (*01, *05, and *06) and L (*02, *08, and *09) alleles (Fig. 5b). Note, WT gl-HEPC74 and gl-HEPC74 $V_H1$-$69$*01 are exactly the same. The gl-AR3C CDRH3 engrafted sequences paired efficiently with the gl-AR3C LC (Supplementary Fig. 5a). Not all gl-HEPC74 engrafted sequences produced sufficient IgG or paired efficiently with gl-HEPC74 LC: $V_H1$-$69$*05, *06, and *08 engraftments were unsuccessful (Supplementary Fig. 5b). The only two gl-HEPC74 CDRH3 engrafted sequences resulting in productive IgG, were based on $V_H1$-$69$-derived L alleles ($V_H1$-$69$*02 and *09).

Of the gl-AR3C CDRH3 engrafted antibodies, only those based on $V_H1$-$69$ F alleles (*01, *05, and the parent allele *06) engaged the E2E1 trimers as efficiently as gl-AR3C itself, while those based on the L alleles (*02, *08, and *09) completely lost binding activity to most E2E1 trimers (Fig. 5b). In contrast, the gl-HEPC74 CDRH3 engrafted onto the $V_H1$-$69$ L alleles retained binding to most E2E1 trimers, except for UKNP6.1.2 E2E1 trimers, which lost binding to gl-HEPC74 engrafted in VH1-69*02 (Fig. 5b).

These observations indicate that the binding of gl-AR3C critically depends on its CDRH3, but that an unsuitable environment, such as the $V_H1$-$69$ L alleles, also can disrupt gl-AR3C binding. In contrast, gl-HEPC74 can accommodate both F and L alleles. Thus, while relying heavily on the CDRH3 loop, the $V_H$ context can affect the activity of AR3C-class precursors. Furthermore, neutralization experiments with isolated CDRH3 peptides derived from mature AR3C-class bNAbs, revealed a lack of neutralizing activity of the peptides alone, consistent with the findings that binding also depends on determinants outside the CDRH3 (Supplementary Fig. 7).

### The G50 polymorphism in $V_H1$-$69$ is critical for binding of inferred germline AR3C and AR3A

It has been postulated that F54 is crucial for binding and neutralization by several AR3C-class antibodies, since the large hydrophobic residue F54 directly interacts with E2[32] and the substitution of this amino acid for an alanine (F54A) leads to complete loss of activity[68]. These results might explain why $V_H1$-$69$ F alleles are preferred over L alleles in AR3C-class bNAbs. However, $V_H1$-$69$ L alleles contain hydrophobic L54 rather than A54. Therefore, to assess the effect of this polymorphism, we generated an F54L substitution (or S54L in case of mature HEPC74) in mature and germline AR3C-class bNAbs containing the CxGGxC motif in the CDRH3 encoding a disulfide bond and evaluated the binding to E2E1 trimers. Surprisingly, the F/S54L substitution did not significantly affect binding of any of the AR3C-class antibodies tested, neither the mature versions nor their inferred germline counterparts (Fig. 6a).

When inspecting the sequences of the tested $V_H1$-$69$ F and L alleles we noted that residue 54 co-varied with residue 50 (Fig. 5a). Thus, F54 was always paired with G50 ($V_H1$-$69$*01, *05, *06), while L54 was always paired with R50 ($V_H1$-$69$*02, *08, and *09). We therefore formulated an alternative hypothesis, i.e., that the arginine at position 50 present in these 'L alleles' is responsible for the loss of gl-AR3C binding. We introduced the G50R mutation in gl-AR3A, gl-AR3C, and gl-HEPC74 alone or in combination with the F54L mutation and tested binding in ELISA. While F54L did not affect binding of gl-AR3A or gl-AR3C (Fig. 6a, b), G50R alone was sufficient to abolish binding of both (Fig. 6b). In contrast, gl-HEPC74 was not affected by G50R nor F54L (Fig. 6b). These findings were reproduced using the mature versions of AR3C and HEPC74 (Supplementary Fig. 5c). Thus, G50R and not F54L in most $V_H1$-$69$ 'L alleles' is responsible for the abolished binding observed when we engrafted the gl-AR3A and gl-AR3C CDRH3s in L-allele $V_H1$-$69$ backbones (Fig. 5a). Note that some more rare $V_H1$-$69$ alleles contain the combination R50 and F54 (*07, *15, *18, and *20) or G50 and L54 (*16). To obtain a molecular understanding of the effect of G50R on gl-AR3A, AR3C, and HEPC74, we used AlphaFold-2.2.0 to generate structural models of gl-AR3A, gl-AR3C, and gl-HEPC74 Fab, either with or without the G50R change[69–71]. We superimposed the predicted models of gl-AR3A, gl-AR3C, and gl-HEPC74 on the available structures of mature AR3A, AR3C and HEPC74 Fabs in complex with E2 (PDB: 6BKB, 4MWF, and 6MEI, respectively) (Fig. 6c). This provided a mechanistic explanation by showing that an arginine at position 50 probably pushes the bent CDRH3 of gl-AR3A and gl-AR3C towards E2 causing a clash, explaining the disruption of its binding, while the straight CDRH3 of gl-HEPC74 is unaffected by the change to arginine (Fig. 6c, Supplementary Fig. 6).

In summary, we show that the G50R but not the F54L polymorphism in the $V_H1$-$69$ alleles can negatively affect the binding of AR3C-class antibodies. However, while this applies to AR3A and AR3C with a bent CDRH3, G50R does not affect binding of HEPC74 with a straight CDRH3. These data suggest that complementary targeting approaches towards a larger group of $V_H1$-$69$-derived precursors including both straight and bent CDRH3 would help overcome the challenge of the F and L alleles.

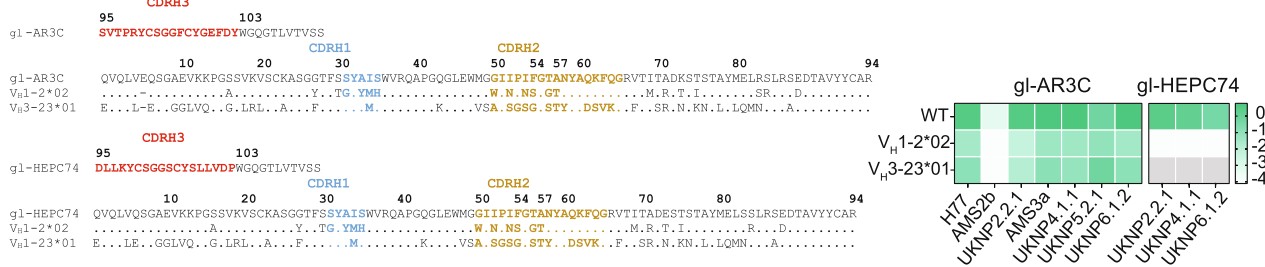

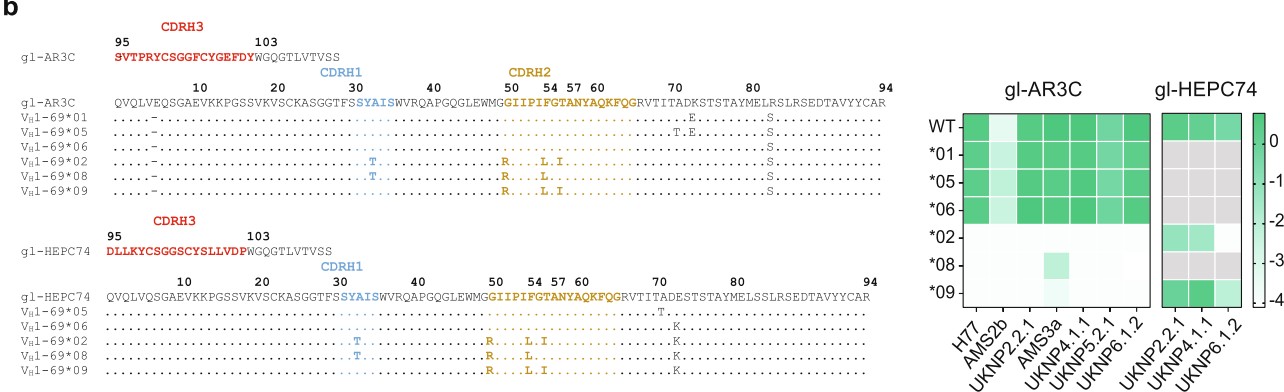

**Fig. 5 | Polymorphisms in $V_H$1-69 alleles impact binding of germline AR3C-class bNAbs. a** Sequence alignment of CDRH3 of gl-AR3C and gl-HECP74 engrafted in $V_H$1-02*02 and $V_H$3-23*01. The CDRH1, 2, and 3 are highlighted. Sequences are numbered using Kabat numbering. Heat map summarizes the ELISA binding of the engrafted antibodies against different E2E1 trimers. Combinations that were not tested because of low production yield, are indicated in gray. **b** Similar to (**a**), sequence alignment of CDRH3 of gl-AR3C and gl-HECP74 engrafted in different $V_H$1-69 alleles. Heat map summarizes ELISA binding results. Combinations that were not tested because of low production yield, are indicated in gray or in the case of gl-HEPC74, $V_H$1-69*01 engrafted sequence is the same as WT sequence.

### Potential AR3C-class precursors are present at high frequency in the human B cell repertoire

An important requirement for germline-targeting strategies is that the pool of naive B cells that are being targeted are present at high enough frequency. To investigate the frequency of $V_H$1-69 in the B cell repertoire, we analyzed a dataset containing the human baseline antibody repertoire of eight healthy individuals[72]. The subject pool was gender-balanced and evenly divided between African-American and Caucasian individuals.

$V_H$1-69-derived B cells constituted ~0.44% of the naive human repertoire, ranging 0.1–0.7% across the eight individuals, amounting to a precursor frequency of approximately 1 in 200 (Fig. 7a). These observations are consistent with findings that most humans carry naive $V_H$1-69 B cells, although frequencies might vary depending on ethnicity[42,67,73]. We then analyzed the frequencies at which we could find F or L alleles in the naive repertoire to take into account this polymorphism in our dataset. We found that F alleles are more predominant at ~0.3% (1 in 300), albeit not found in one of the eight individuals. The L alleles, on the other hand, were found amongst all individuals but at a lower frequency of ~0.13% (1 in 800). When querying the naive $V_H$1-69-derived B cells, including both F and L alleles, that also had a longer than usual CDRH3 loop (i.e., longer than 19 a.a.), the precursor frequency decreased to around 1 in 1000 (~0.107%) (Fig. 7a).

We further refined our analysis by considering the CxGGxC disulfide bond motif in the CDRH3 that is common to many, but not all, AR3C-class bNAbs[43,74]. The frequency of cells that utilize $V_H$1-69, contain an elongated CDRH3 (>19 amino acids) and display the CxGGxC motif in the CDRH3 is on average 1 in 12,500 cells (~0.008%) (Fig. 7a). This frequency is ~30-fold higher than the proposed precursor frequency of VRC01-class B cells (1 in 400,000)[25], suggesting that these

naive $V_H$1-69 B cells are feasible targets for germline-targeting HCV vaccine strategies.

Next, we compared the $V_H$ gene expression in the complete and in the naive B cell repertoire (Fig. 7b). In contrast to the majority of genes, the frequency of $V_H$1-69 gene usage substantially increased from around ~0.44% in the naive to 4.2% in the complete B cell repertoire (~10-fold change) (Fig. 7b, c). This strong enrichment suggests that relatively rare $V_H$1-69 naive B cells are activated very efficiently during life, possibly by viral infections such as seasonal influenza[55]. Furthermore, the $V_H$1-69 gene has been found to be overrepresented in HCV-infected patients when compared to healthy donors, indicating the capacity of B cells utilizing this gene to be involved in an anti-HCV immune response[75].

The amino acid sequences of precursor and mature HCs of $V_H$1-69 derived AR3C-class bNAbs share 80% identity on average (range: 73% to 90%) (Fig. 7d). This is significantly higher than $V_H$1-2-derived VRC01-class HIV-1 bNAbs, which only share 60% (range: 55%-69%) amino acid identity with their respective germline HC precursor (Fig. 7d)[29,65,76–80]. Despite this low sequence identity, germline VRC01-class B cells can be guided to evolve into mature bNAbs using an intricate sequential vaccine regimen in certain mouse models[81]. The relatively low level SHM of AR3C-class bNAbs suggests that shepherding germline precursors of AR3C-class should be a more achievable goal, and the elicitation of such a response might not require an overly complex sequential vaccination regimen.

### Discussion

In this study, we identified HCV E2E1 trimers that bind to AR3C-class germline antibodies. We further show that selected E2E1 trimers assembled into nanoparticles activate inferred germline B cells in vitro. Exploiting these E1E2 trimers, we characterized two subclasses of

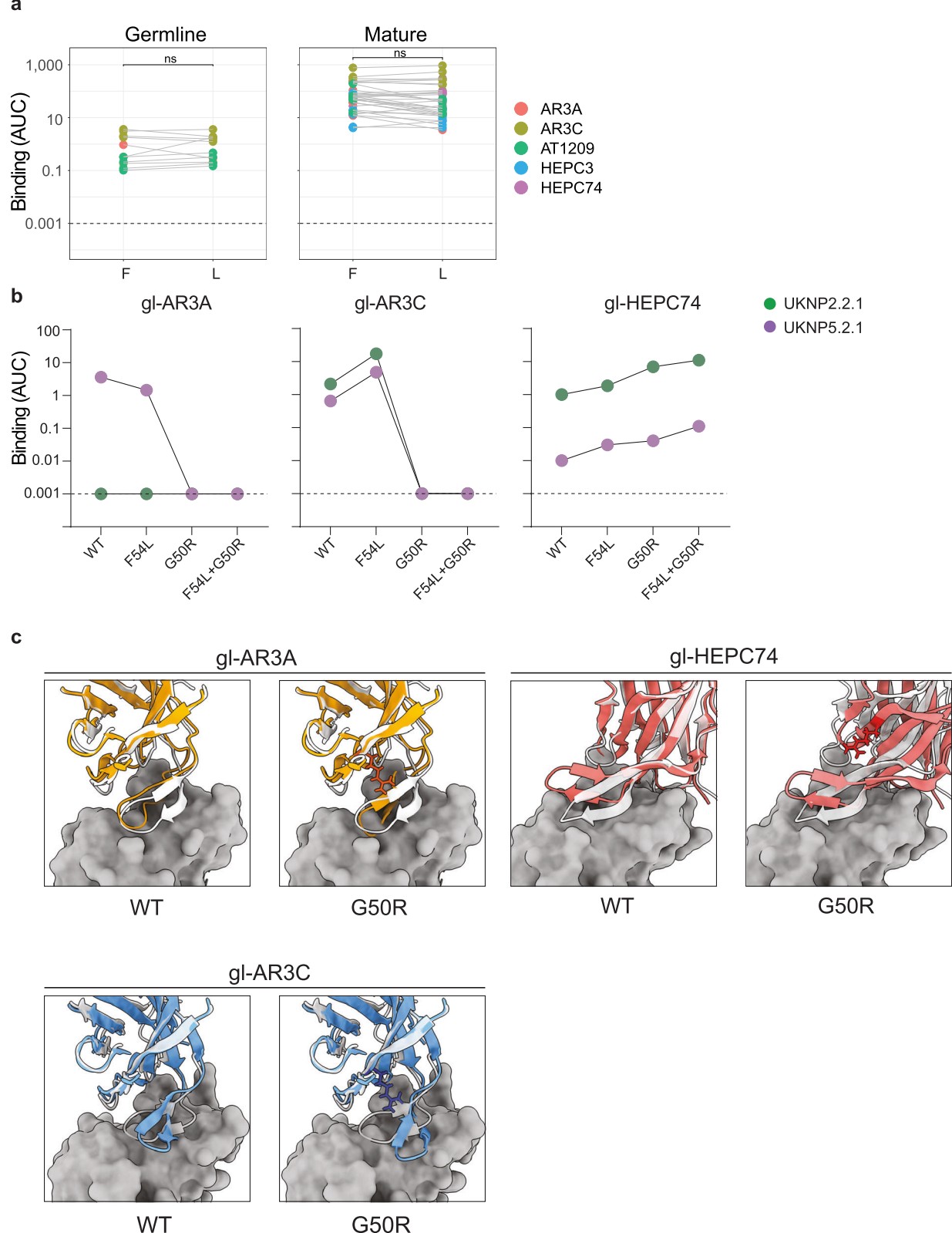

AR3C-class HCV bNAbs (i.e., those with bent and those with straight CDRH3) and their germline precursors and determined sequence features that are critical for binding for each of the subclasses as determined by gl-AR3A/gl-AR3C and gl-HEPC74 results.

Inferred germline antibodies are widely used to design B-cell-lineage immunogens in vaccine development[82]. Despite being a commonly used tool to guide immunogen design, inferred germline antibodies represent a prediction of the naive B cell receptor and follow-up studies are needed to determine if our immunogens are able to engage the desired $V_H1$-69-derived naive B cells. Importantly, in a previous study we showed that an E1E2 trimer based on the H77 reference strain failed to engage naive B cells isolated from healthy donors[42]. Here, we found that that same H77-derived E2E1 weakly engaged gl-AR3C and gl-HEPC74 and failed to

**Fig. 6 | Study of the F54 and G50 polymorphisms in binding of selected AR3C-class bNAbs. a** Binding of AR3C-class antibodies displaying the CxGGxC motif (left, germline; right, mature) with either F54 (F) or L54 (L) to all E2E1-I53-50A trimers used in this work. Note that mature HEPC74, contains S54 instead of F54 and is mutated here to S54L. A two-sided Wilcoxon matched-pairs signed-rank test was used for the comparison (ns: non-significant). All experiments were performed twice independently. **b** ELISA binding of gl-AR3A, gl-AR3C and gl-HEPC74 or the same mAbs with CDRH2 mutations F54L, G50R or F54L + G50R to UKNP2.2.1 and UKNP5.2.1 E2E1 trimers. Binding is represented as AUC and the dotted line represents the detection limit. **c** Top left panels: Alphafold2 predicted structure of gl-AR3A or gl-AR3A G50R (in orange) superimposed on the available structure of mature AR3A (white) in complex with of E2 (dark gray) (PDB: 6BKB). Bottom left panels: Alphafold2 predicted structure of gl-AR3C or gl-AR3C G50R (in blue) superimposed on the available structure of mature AR3C (white) in complex with of E2 (dark gray) (PDB: 4MWF). Top right panels: Alphafold2 predicted structure of gl-HEPC74 or gl-HEPC74 G50R (red) superimposed on the available structure of mature HEPC74 (white) with E2 (dark gray) (PDB: 6MEH). For all G50R panels, the arginine at position 50 mutation is highlighted in sticks.

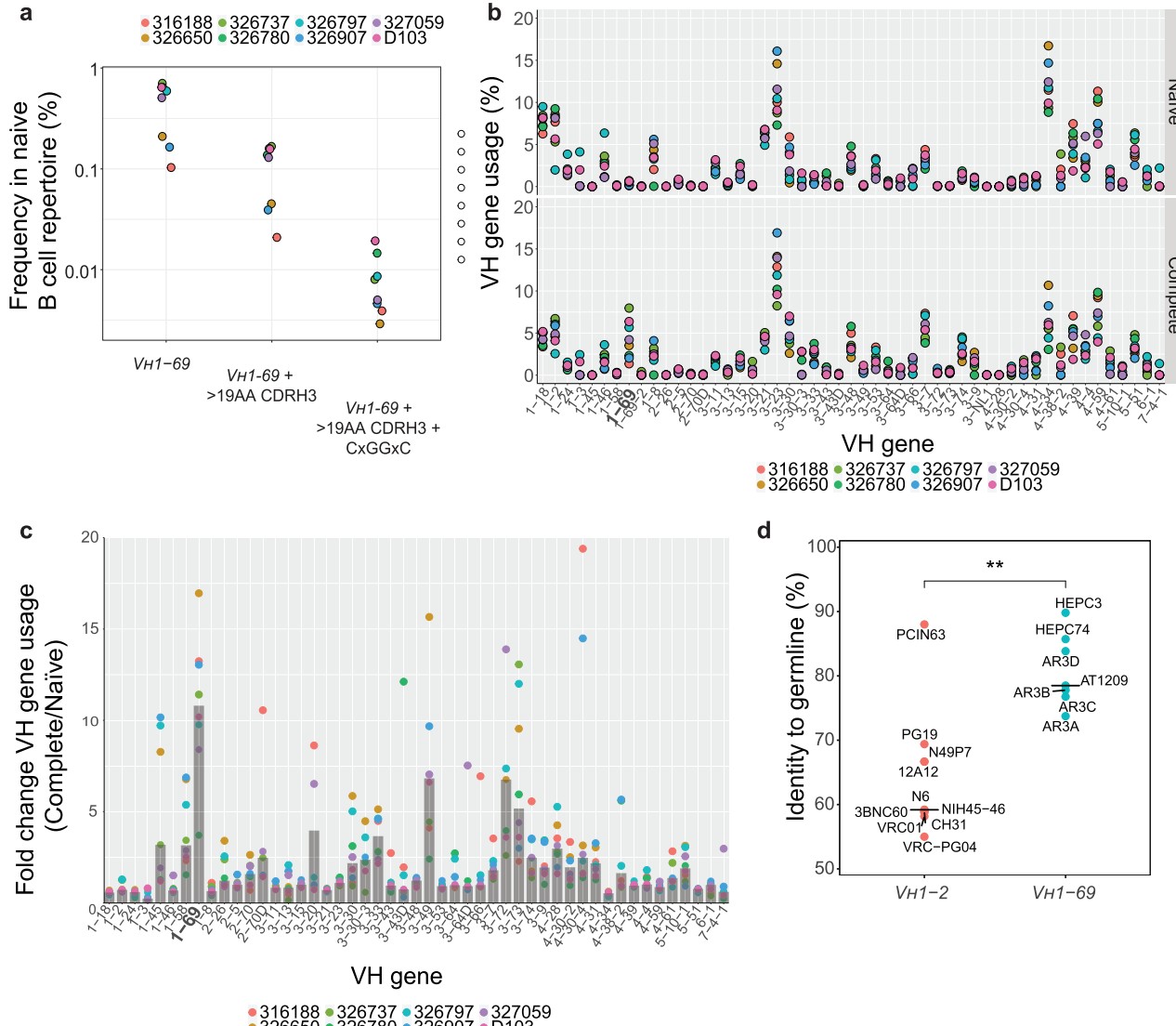

**Fig. 7 | In silico analysis of the predicted frequency of AR3C-class bNAb precursors in healthy human individuals. a** AR3C-class bNAb precursors in the naive B cell repertoire. The frequencies at which the shared characteristics of AR3C-class bNAb precursors are found in the naive B cell repertoire (data available at https://github.com/briney/grp_paper[72]) are shown for the different donors: all $V_H1$-$69$ (left), $V_H1$-$69$ combined with a long CDRH3 (>19 amino acids, middle), $V_H1$-$69$ with a long CDRH3 and CxGGxC motif (right). **b** Comparison of the $V_H$ gene usage in the complete and naive B cell repertoires of eight individuals. $V_H$ gene usage was calculated as a percentage of the naive (upper panel) or complete (bottom panel) B cell repertoire. **c** $V_H$ gene usage in the complete B cell repertoire represented as the fold change compared to the naive repertoire calculated for each of the eight individuals. Gray bars represent the median values per $V_H$ gene. **d** Comparison of levels of somatic hypermutation (SHM) between the 7 $V_H1$-$69$-derived AR3C-class mature HCV bNAbs and 10 $V_H1$-$02$-derived VRC01-class HIV-1 bNAbs at the amino acid level. The crossbars represent median percentage identity to germline. A two-sided Mann-Whitney U test was used for the comparison (**P = 0.0054).

activate the corresponding B cells, and did not bind to other AR3C-class gl-bNAbs. These findings highlight the need for rationally selected E1E2 and/or optimized E1E2 designs in order to improve the desired binding capabilities that are capable of engaging a large pool of potential naive B cell precursors with high(er) affinities. Next steps can include improving the E2E1 immunogen, using the UKNP4.1.1 sequence as the basis, in order to bind a broader pool of $V_H1$-$69$ germline precursors and then evaluate whether it binds to the desired naive human $V_H1$-$69$ B cells in the B cell repertoire of healthy human individuals. Such ex vivo experiments can then inform whether the designed germline-targeting immunogens should be evaluated in human clinical trials[25,83].

Our data confirm that AR3C-class antibodies, especially in the case of gl-AR3C and to a certain extend for gl-HEPC74, and their precursors engage E1E2 utilizing mostly their heavy chain and showed that the CDRH3 is crucial but not exclusive for this engagement as soluble CDRH3 peptides failed to neutralize HCVpp's (Supplementary Fig. 7). We showed that the R50 polymorphism and not L54 in the CDRH2 of L allelic VH1-69 results in loss of binding of gl-AR3A and gl-AR3C, with a bent CDRH3 bNAb, while gl-HEPC74, with a straight CDRH3, was unaffected. These polymorphisms highlight that complementary immunogens are needed to target different AR3C-class precursors that can accommodate the different $V_H1$-69 alleles. Another consideration is that a recent study using influenza spike nanoparticle immunogens suggests that $V_H1$-69 L alleles are selected against due to autoreactivity caused by the L on 54[84]. Whether this is the case for HCV bNAbs needs to be investigated as this would have repercussions for the wide-spread use of a germline-targeting $V_H1$-69 vaccine.

A recent study showed that human AR3C-class HCV bNAbs can employ an alternative binding mode in which the CDRH2-loop rather than the CDRH3-loop plays the most critical role in binding, by engaging the front layer of E2 near Cys429, and in which the LCs make substantial more contacts[75]. The same study also highlighted the relevance of CDRH1 and CDRH2 in the maturation of these bNAbs. These new insights should be taken into account when designing germline-targeting vaccination approaches for this class of bNAbs[75].

Most HIV-1 immunogens based on sequences from circulating HIV-1 strains lack detectable binding for the predicted germline precursors of bNAbs[26,85–88]. In contrast, selected HCV E1E2 proteins, especially the one derived from UKNP4.1.1 strain, are able to engage a number of germline bNAb precursors without requiring designed epitope changes. However, germline-targeting immunogens should ideally target a pool of diverse bNAb precursors. Therefore, it is prudent to optimize candidate E2E1 to increase affinity to germline bNAbs and increase the pool of potential germline precursors that can be engaged. For example, removing glycosylation sites on E1 and E2 can increase recognition of some AR3C-class bNAbs[89–91] and altering glycosylation patterns using different production cell lines can improve E2 immunogenicity and antigenicity[46,92]. Furthermore, the removal of the hypervariable region 1 (HVR1) on E2 should improve exposure of AR3, which might help engagement of germline AR3C- bNAbs[93–95]. In addition, the conformational flexibility of the CD81bs which is part of AR3 poses another potential hurdle for immune recognition[10] and structure-based stabilization of AR3 could be another avenue to improve germline AR3C-class bNAbs engagement. In addition, to increase avidity, nanoparticle display is beneficial, especially for lower affinity interactions such as those of bNAb precursors with a candidate immunogen[85]. The two-component nanoparticle platform used here has the added advantage of enabling mosaic display of different strains of HCV trimers on the same nanoparticle to direct responses to shared epitopes[45]. Lastly, the recently resolved E1E2 cryo-EM structure and an in-depth study on functional residues in the E1E2 glycoprotein will also facilitate the design of new HCV vaccine immunogens[23,96].

To summarize, by identifying E1E2 proteins capable of binding AR3C-class precursors and by further defining the signatures of AR3C-class precursors, we provide a framework for the design of potent germline-targeting vaccines for HCV. We hypothesize that few modifications to the immunogens are required to obtain a priming vaccine candidate and that a vaccine regimen aimed at eliciting HCV AR3C-class bNAbs might not require an overly long and complex vaccination regimen. We propose UKNP4.1.1 E2E1 trimer as a promising template for generating an immunogen with broad and high-affinity binding to germline AR3C-class antibodies.

## Methods

### Constructs
All antibody sequences were human codon-optimized and cloned into human IgG1 expression vectors for the corresponding HCs or LCs as described before[51,97,98] (Genscript Biotech). Antibody amino acid residues are numbered according to the Kabat convention[99]. Most mature and germline inferred antibody sequences were retrieved from previously published[32,43]. The LC of HEPC3 and HEPC74 were inferred using IMGT/V-QUEST software tool[44]. The inferred germline AT1209 sequences were a kind gift of Tim Beaumont and Sabrina Merat (AIMM therapeutics).

HCV E1E2 amino acids are numbered according to the standard H77 polyprotein numbering (GenBank AF009606)[100]. The panel of seven E2E1-I53-50A constructs was described elsewhere[45]. In short, E1E2 sequences were permutated (i.e., E2E1) by fusing the E2 ectodomains (384-698) to the N-terminus of the E1 ectodomain (192-325), separated by a furin site to ensure proper cleavage between E2 and E1. The I53-50A component[101] was genetically fused to C-terminus of E1 to trimerize E2E1 and allowing nanoparticle assembly[40,45,101,102]. All constructs contain a strepII-tag to facilitate purification.

### Antibody expression and purification
All antibodies were transiently expressed in HEK293F cells similarly to the proteins. Suspension HEK293F cells (Invitrogen, cat no. R79009) at a density of 0.8–1.2 million cells/mL were co-transfected with two plasmids expressing IgG1 heavy chain (HC) and light chain (LC) in a 1:1 ratio using 1 mg/mL PEI MAX (Polysciences) (1:3 plasmid:PEI Max). Supernatants containing the antibodies were harvested five days post-transfection, centrifuged for 30 min at $4000 \times g$ and filtered using 0.22 µm Steritop filters (Merck Millipore). The filtered supernatant was run over a 1 mL protein A/G column (Pierce) followed by two column volumes of PBS wash. The antibodies were eluted with 18 mL 0.1 M glycine pH 2.5 directly captured in 2 mL neutralization buffer (1 M TRIS pH 8.7). The purified antibodies were buffer exchanged to PBS using 100 kDa VivaSpin20 columns (Sartorius). The IgG concentration was determined using a NanoDrop 2000 and the antibodies were stored at 4 °C until further analyses.

### Protein expression and purification
StrepII-tagged E2E1-I53-50A trimers were produced in HEK293F (Invitrogen, cat no. R79009) cells maintained in Freestyle medium (Life Technologies). Cells were co-transfected at a density of 0.8–1.2 million cells/mL by addition of a mix of PEImax (1 µg/µl) with expression plasmids (312.5 µg DNA/L of 293F cells) in a 1:1 ratio (trimer:furin) in OptiMEM. Supernatants were harvested six days post transfection, centrifuged for 30 min at $4000 \times g$ and filtered using 0.22 µm Steritop filters (Merck Millipore). The glycoproteins were purified using StrepTactinXT columns (IBA Life Sciences) by gravity flow (~0.5–1.0 ml supernatant/min). For flow-cytometry experiments (see below), we generated AviHis-tagged E2E1-foldon trimers in which the strepII-tagged I53-50A was replaced by foldon trimerization domain with a C-terminal AviHis-tag. E2E1-foldon-AviHis proteins were purified from 293F supernatant using NiNTA agarose beads (Qiagen) and polished using a Superdex 200 Increase column in PBS. The eluted proteins were concentrated and buffer exchanged into 5% glycerol Tris-buffered saline (TBS, pH 7.5) using Vivaspin 6, 100,000 MWCO PES (Sartorius). Finally, purified proteins were fractionated using size-exclusion chromatography (SEC) on a Superose 6 Increase GL (GE Healthcare) and the fractions corresponding to trimer were pooled and concentrated. The concentration was determined by Nanodrop (Thermo Fisher Scientific) using theoretical molecular weight and extinction coefficient. After purification, avi-tagged proteins were biotinylated with a BirA500 biotin-ligase reaction kit according to the manufacturer's instruction (Avidity).

## E2E1-NP assembly

E2E1-I53-50 nanoparticles (E2E1-NP) were assembled essentially as described (Brouwer 2019; Sliepen submitted). E2E1-I53-50A trimers were mixed in an equimolar ratio with I53-50B.4PT1 (kind gift from Neil King and Rashmi Ravichandran) for an overnight (-16 h) incubation at 4 °C. The assembly mix was then concentrated at 1000 g using Vivaspin filters with a 10 kDa molecular weight cutoff (Sartorius) and passed through a Superose 6 Increase column in TBS/5% glycerol, pH 7.5. The fractions corresponding to the assembled NPs were pooled and concentrated at 500 g using Vivaspin filters with a 10 kDa molecular weight cutoff. Nanoparticle concentrations were determined with a Nanodrop using the peptidic molecular weight and extinction coefficient.

## SDS-PAGE

SDS-PAGE analyses were performed as described elsewhere[103] with some modifications. Briefly, 5 μg of E2E1 trimer were mixed with loading dye (25 mM Tris, 192 mM Glycine, 20% v/v glycerol, 4% m/v SDS, 0.1% v/v bromophenol blue in milli-Q water), and incubated at 95 °C for 10 min prior to loading on a 10–20% Tris-Glycine gel (Invitrogen). For reducing SDS-PAGE, dithiothreitol (DTT; 100 mM) was included in the loading dye and loaded on a Novex 10–20% Tris-Glycine gel (Thermo Fisher Scientific). Gels were run in a buffer containing 25 mM Tris, 192 mM glycine, and 0.5% SDS for 1 h at 200 V at 4 °C. Coomassie blue staining of SDS-PAGE gels was performed using the PageBlue Protein Staining Solution (Thermo Fisher Scientific).

## StreptactinXT ELISA

Purified strepII-tagged E2E1 trimers (2.1 μg/mL in TBS) were coated for 2 h at room temperature on 96-well Strep-TactinXT coated microplates (IBA LifeSciences). Plates were washed with TBS twice before incubating with serially diluted mAbs in casein blocking buffer (Thermo Fisher Scientific) for 90 min. After three washes with TBS, a 1:3000 dilution of HRP-labeled goat anti-human IgG (Jackson Immunoresearch) in casein blocking buffer was added for 45 min. After washing the plates five times with TBS + 0.05% Tween-20, plates were developed by adding develop solution [1% 3,3′,5,5′-tetraethylbenzidine (Sigma-Aldrich), 0.01% $H_2O_2$, 100 mM sodium acetate, 100 mM citric acid] and the reaction was stopped after 3 min by adding 0.8 M $H_2SO_4$. Absorbance was measured at 450 nm and area under the curve (AUC) values were calculated using Graphpad Prism. All data was normalized t against the binding signal of the AP33 bNAb, which was used as loading control[104].

BLI assays were performed using an Octet K2 instrument (FortéBio). All assays were performed at 30 °C and with agitation speed of 1000 rpm. Antibodies and E2E1 samples were dissolved in running buffer (PBS, 0.02% Tween, 0.1% bovine serum albumin (BSA)) in a volume of 250 μl/well. Antibodies (1 μg/mL) were immobilized onto protein A biosensors (FortéBio, cat no. 18–5010) until a loading threshold of 1.0 nm was reached, followed by a 30 s baseline measurement in the running buffer. Purified E2E1 trimers were diluted to 500 nM and association and dissociation were measured for 300 s each. A well containing running buffer without protein was used for background correction. Data was analyzed and visualized in GraphPad Prism 8.3.0.

## HCVpp production

HCV pseudoparticle (HCVpp) generation and neutralization assays were performed as described elsewhere[105]. One day prior to transfection for generating HCVpp, $1.5 \times 10^6$ HEK293T or HEK293T[CD81KO] cells were seeded on a 10 cm² dish. Cells were co-transfected with three plasmids: MLV Gag-Pol packaging construct, firefly luciferase[106] and E1E2 in optimized ratios[105] with a total amount of 6 μg of DNA and 12 μl of Lipofectamine 2000 (Invitrogen) in Opti-MEM (ThermoFisher). After an incubation overnight, Opti-MEM was replaced by DMEM (Gibco)/10% fetal bovine serum (FCS)/0.1% Penicillin-Streptomycin (PS). Two days later, the supernatant containing the HCVpps was passed through a 0.45 μm filter and frozen at −80 °C for long-term storage or 4 °C when used within a week.

## Neutralization assays

Huh-7 cells, a gift from François-Loïc Cosset, a hepatocyte-derived carcinoma cell line) were seeded at $1.5 \times 10^4$ cells per well in a 96-well plate 24 h prior to the experiment in 100 μl DMEM/10% FCS/0.1% PS/1% nonessential amino acids/1% HEPES buffer (Huh-7 medium). HCVpps were incubated with serially diluted mAb concentration in duplicate at 37 °C in 5% $CO_2$. After 1 h, the HCVpp/mAb mixture (30 μL) was added to the Huh-7 cells and incubated for 4 h. After the incubation, 220 μl Huh-7 medium and incubated for 72 h. After removing the media, cells were lysed and luciferase signal was measured using the Luciferase Assay System (Promega) and a GloMax luminometer (Promega, USA).

## Analysis of human BCR repertoire

All data on the human BCR repertoire can be found at https://github.com/briney/grp_paper[72]. The datasets were analyzed in RStudio 1.2.5033 (R4.0.4). Because of memory capacity limitations, data from 8 out of the 10 subjects were used. Naive BCR sequences were defined as having no amino acid mutations respective to the germline precursor and were extracted from each donor dataset individually.

## Statistical analysis

Statistical analyses on neutralization titers ($IC_{50}$ values) and ELISA results (AUC), as well as data fitting for ELISA were performed using GraphPad Prism 8.3.0. RStudio 1.2.5033 (R4.0.4) was used for all other statistical analyses and data visualization. See figure legends for details on the accompanying analyses.

## Structure prediction and visualization

Molecular graphics and analyses were performed with UCSF ChimeraX, developed by the Resource for Biocomputing, Visualization, and Informatics at the University of California, San Francisco, with support from National Institutes of Health R01-GM129325 and the Office of Cyber Infrastructure and Computational Biology, National Institute of Allergy and Infectious Diseases[107,108]. For predicting the structures of (mutant versions of) gl_AR3C and gl-HEPC74 Fabs AlphaFold-2.2.0 Multimer was used[69–71]. Code available at: https://github.com/deepmind/alphafold.

## Generation of AR3C- and HEPC74-expressing B cells

HCV-specific Ramos B cells stably expressing gl-AR3C, AR3C, gl-HEPC74, or HEPC74 BCRs were generated as described elsewhere[52]. In short, the 2-1261gl gene of the pRRL EuB29 2-1261gl IgG TM.BCR.GFP.WPRE plasmid[109] was exchanged for the heavy and light chain genes of either gl-AR3C, AR3C, gl-HEPC74, or HEPC74 using Gibson assembly (Integrated DNA Technologies). Lentiviruses were produced by co-transfecting the generated expression plasmid with pVSV-g, pMDL, and pRSV-Rev into HEK293T cells using lipofectamine 2000 (Invitrogen). Two days post-transfection, HEK293T supernatant was used to transduce IgM-negative Ramos B. Seven days post-transduction, BCR-expressing B cells were FACS sorted on GFP and IgG double-positivity (i.e., BCR-expressing cells) using a FACS Aria-II SORP (BD Biosciences). B cells were expanded and cultured indefinitely. Prior to B cell binding assays and calcium flux assays, cells were selected for GFP expression to ensure similar levels if IgG (as BCR) across the cell lines (Supplemental Fig. 3b).

## Ramos B cell binding assay

Antigen specificity of (gl-)AR3C and (gl-)HEPC74 Ramos B cells to H77 E2E1 and UKNP4.1.1 E2E1 was detected using labeled probes and flow cytometry as previously[42,52]. Briefly, biotinylated H77 and UKNP4.1.1 E2E1-foldon-Avi-His were individually multimerized with fluorescently

labeled AF647 (Biolegend) and BV421 (Biolegend) streptavidin at a 2:1 protein to fluorochrome molar ratio and incubated for 1 h at 4 °C. Unbound streptavidin conjugates were quenched with 10 μM biotin (Genecopoiea) for 15 min. The antigen-probe cocktails were then used to stain $5 \times 10^5$ cells together with live/DEAD dye (eBioscience™ Fixable Viability Dye eFluor™ 780, Thermo Fisher), IgG PE-Cy7 (G18-145, BD Biosciences) in FACS buffer (PBS supplemented with 1 mM EDTA and 2% fetal calf serum). The live/DEAD marker was titrated for signal-to-noise ratio and IgG PE-Cy7 was used in a dilution of 2.5 μL in 50 μL. Stained samples were subsequently washed twice with FACS buffer and acquired on the BD LSRFortessa™ for cell analysis. Analysis was performed using FlowJo v10.8.1. Ramos cells were first gated based on the morphology (FSC-A/SSC-A) and doublets were removed. Live cells were selected and subsequently gated on GFP+ and IgG+. Antigen-specific Ramos B cells were double positive for the HCV H77 and 4.1.1 E2E1- foldon-Avi-His probes (AF647 and BV421).

### B cell activation assay

B cell activation experiments of the generated HCV-specific B cells were performed as previously described[52]. In short, 4 million cells/mL in RPMI+ + were loaded with 1.5 μM of the calcium indicator Indo-1 (Invitrogen) for 30 min at 37 °C, washed with Hank's Balance Salt Solution supplemented with 2 mM CaCl$_2$ and 0.5% FCS, followed by another incubation of 30 min at 37 °C.

Antigen-induced Ca$^{2+}$ influx of AR3C(gl) and HEPC74(gl) B cells was assessed on a LSR Fortessa (BD Biosciences) by measuring the 379/450 nm emission ratio of Indo-1 fluorescence upon UV excitation. Following 30 s of baseline measurement, aliquots of 1 million cells/mL were then stimulated for 250 s at RT with either H77 E2E1-I53-50A trimer, H77 E2E1-I53-50NP, UKNP4.1.1 E2E1-I53-50A trimer or UKNP4.1.1 I53-50NP. For the stimulation, trimers and nanoparticles were compared using equimolar amount of the E2E1-I53-50A component. Maximum Indo-1-fluorescence was determined by adding ionomycin (Invitrogen) to a final concentration of 1 mg/mL. Kinetics analyses were performed using FlowJo v8.1.

### Reporting summary

Further information on research design is available in the Nature Portfolio Reporting Summary linked to this article.

## Data availability

All data on the human BCR repertoire can be found at https://github.com/briney/grp_paper[72]. Designed constructs are based on sequences deposited at GenBank: H77 (AAB67037; mutations R564C, V566A, G650E), UKNP2.1.1 (KU285220.1), AMS2b (KR094963.1), AMS3a (KR094964.1), UKNP4.1.1 (ALV85530.1), UKNP5.2.1 (ALV85536.1), and UKNP6.1.2 (ALV85538.1). Structures of AR3C-class bNAbs AR3C, AR3A, AR3B, AR3D, HEPC3, HEPC74, and AT1209 are publicly available (PDB entries: 4MWF, 6BKB, 6BKC 6BKD 6MEI, 6MEH, 7T6X, respectively) as well as the E1E2 complex (PDB: 7T6X). Other data that support the findings of this study are available from the corresponding authors (K.S. and R.W.S.) upon reasonable request. Source data are provided with this paper.

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

## Acknowledgements

We thank Tim Beaumont and Sabrina Merat (AIMM) for providing sequences and data on AT1209. We thank Andrew McGuire for kindly sharing the pRRL.EuB29 lentiviral vector that was used to transduce Ramos B cells. The following reagent was obtained through the NIH AIDS Reagent Program, Division of AIDS, NIAID, NIH: Ramos B cells from Drs. Li Wu and Vineet N. KewaliRaman. We thank Rashmi Ravichandran and Neil P King for providing the I53-50B.4PT1 protein. This research was supported by the Fondation Dormeur, Vaduz (to R.W.S and M.J.v.G), an

AMC Fellowship from Amsterdam UMC (M.J.v.G.), a Vici grant from the Netherlands Organization for Scientific Research (NWO) (R.W.S.), a Vidi and Aspasia grant from the NWO (grant numbers 91719372 and 015.015.042) (J.S.), the Bill & Melinda Gates Foundation (OPP1156262 to R.W.S.), an Amsterdam institute for Infection and Immunity Postdoctoral grant (K.S.) and an AMC PhD Scholarship (A.C.M.). L.R. was funded by the AMC as part of the local scientific research incentive policy.

## Author contributions

Conceptualization: J.C.P., R.W.S., and K.S. Funding acquisition: K.S., R.W.S., M.J.v.G., and J.S. Investigation: J.C.P., M.d.G., L.R., I.Z., A.C., S.K., and W.O. Methodology: J.C.P., M.d.G., L.R., M.J.v.G., R.W.S., and K.S. Project administration: J.C.P., R.W.S., and K.S. Supervision: J.C.P., M.J.v.G., J.S., R.W.S., and K.S. Writing—original draft: J.C.P., R.W.S., and K.S. Writing—review & editing: all authors.

## Competing interests

The authors declare no competing interests.
