## [Peer Review File · Nature Communications]

REVIEWER COMMENTS

Reviewer #1 (Remarks to the Author):

In this manuscript, the authors investigate the suitability of naive B cells expressing germline AR3C-class B cell receptors as vaccine targets for a germline-targeting approach. They show that AR3C-class germline antibodies are able to bind (and in some cases neutralize) some E2E1 trimers (candidate immunogens). They further show that a B cell line bearing germline AR3C-class B cell receptors can become activated by some E2E1 trimers (as measured by calcium influx). Lastly, naive B cells with AR3C-like B cell receptors (VH1-69, long CDRH3, +/- CxGGxC motif) are found in healthy human subjects. This work lays useful groundwork for HCV vaccine development.

The authors also undertake a detailed characterization of the features of AR3C-class mature and germline antibodies and how those features affect antigen binding. This includes the influence of the light chain and VH1-69 restriction on E1E2 binding as well as analysis of allelic variation in VH1-69 sequence. Structural data/modeling is used to suggest mechanistic explanations for their findings (e.g., bent vs straight CDRH3 in complex with E1E2). These data are provocative but ultimately, since the work only includes 2 antibodies (and their germline equivalents), the authors' current conclusions are not supported.

Major issues:

1. The authors' claims about AR3C-class antibodies are overstated. For their analyses, they tested germline and mature variants of AR3C and HEPC74 only and it is unclear to what extent these findings are generalizable to all AR3C-class antibodies. This is especially true given that the results for AR3C and HEPC74 often differ from one another (e.g., VH1-69 dependence/allelic variation and light chain dependence to a lesser degree). Furthermore, these 2 antibodies are taken as representatives of "bent" and "straight"-type AR3C-class antibodies but again, with only 1 antibody per group, no generalizable conclusions can be drawn. The authors would ideally expand their analysis to include more AR3C-class antibodies, but if this is not tenable then, at a minimum, conclusions for this portion of the manuscript should be significantly more limited.

2. When characterizing how various antibody features affect function, binding, but not neutralization, is tested. Ultimately, the goal is that individuals will produce AR3C-like mature antibodies that are broadly neutralizing. Therefore, it seems an oversight not to show the affect that different VH or light chains have on neutralization.

Minor issues

1. The sequence data in fig 5a-b is very difficult to read. Similarly the VH data in fig 6b-c is hard to parse. Please increase the size of these figures.

Reviewer #2 (Remarks to the Author):

SUMMARY

This is a follow up to a recently published manuscript in Nature Comm by Sliepen et al where they described novel immunogenic permuted E1/E2 immunogenic HCV nanoparticles that induced cross neutralizing antibodies upon immunization in rabbits. Here, they utilize the same constructs to examine their capacity to bind germline precursors of known HCV broadly neutralizing antibodies (bNAbs). The authors focus on the AR3C inferred VH1-69 germline precursors of AR3C-class bNAbs as these were the ones that showed the highest binding. They further characterize this binding and interactions. They show that these constructs can bind and activate B cells expressing inferred germline AR3C-class bNAbs precursors as B cell receptors but only at the highest concentrations. They show that VH1-69 allelic variation affects germline AR3C binding. They also examine a dataset containing the human baseline antibody repertoire of eight healthy individuals and show that potential AR3C-class precursors are present at high frequency in the human B cell repertoire.

COMMENTS

The manuscript provides a novel and interesting follow-up on their previous paper and provides compelling evidence that these constructs bind and activate inferred germline precursors of the AR3C broadly neutralizing antibodies. The interactions are well characterized which are key for further development of these constructs for vaccination. I have the following comments:

1. The constructs only activate germline precursor expressing B cells at the highest concentration. It is also clear from Supplementary Figure 3 that the expression of the gl precursors of HEPC74 is very low. Please discuss on how this affects the overall activation of B cells.
2. Figure 5C examines all the antibodies produced but we cannot distinguish which is which. Please represent as different colours. Also, the legend mentions top and bottom panels which should be right or left.

3. There is yet no proof regarding the ability of these constructs to stimulate germline precursors in vivo in a suitable model that would be relevant to humans. This could be challenging, please discuss.

MINOR COMMENTS:

There are multiple typos in the document. Please revise carefully.

Reviewer #3 (Remarks to the Author):

“Germline-targeting” is an interesting and timely strategy for vaccine design. In this manuscript, the authors generate a panel of VH1-69-derived, AR3C-class bnAb precursors to investigate HCV immunogens capable of binding the precursors and the signatures in the precursors for binding. They detect cross-binding of their E2E1 glycoprotein trimers to AR3C and HEPC74 precursors and activation of B cells expressing these precursors by a high concentration of selected E2E1 nanoparticles, but not trimers. The authors also assessed the genetic and amino acid sequence features that affect precursor binding which include an extended CDRH3 and VH1-69 allelic polymorphism. At the end of the abstract, the authors stated the identification of critical signatures in two subclasses of the AR3C-class bNAbs that will allow refined protein design to provide a framework for germline-targeting vaccine design.

A number of areas require clarification:

1. The bNAbs and their precursors studied have many features and their major molecular and structural features have already been reported previously. What specific signatures reported in this study are really important in germline-targeting vaccine design, and how to implement it? The authors showed their to-be-published nanoparticles can bind and stimulate the precursors, but this should apply to any nanoparticles that present E2 multivalently. How do the results impact germline-targeting and vaccine design better than what is already known in the field?

2. The study only analyzed signatures in the AR3C and HEPC74 precursors. The data show that the two antibodies are quite different in sequence (CDRH1, 2 and 3) and binding requirements (heavy chain dependence, VH1-69 background and allelic polymorphism) (Figures 3-5). There seems to be no common signatures in the two antibody precursors and how these two antibodies represent a class of VH1-69 bnAbs with a bent or a straight CDRH3, respectively. Therefore in their conclusion, the features that are critical for binding may not be applied to other VH1-69 bnAbs.

3. Any explanation why some germline bnAb precursors displayed neutralizing activity in the absence of detectable binding to the soluble trimer (Supplementary Figures 1 &2). Any detectable difference between the env glycoprotein trimer and env on HCVpp?

4. Any explanation why the germline AR3C engrafted on VH1-2*02 and VH3-23*02 (which only has 79% and 57% sequence identity with VH1-69*01, respectively) retained binding capacity to all E2E1 trimers tested (except AMS2b), but no binding when engrafted on VH1-69 L alleles (>95% sequence identity) (Figures 5a and 5b)?

5. In the heat map summarizing ELISA binding results in Figure 5, what are the WT for gl-AR3C and gl-HEPC74, respectively? It seems AR3C is derived from VH1-69*06 and HEPC74 from VH1-69*01, respectively. However in this figure, binding of AMS2b to gl-AR3C WT and to *06 is quite different. Similarly, gl-HEPC74 WT show moderate binding to all the E2E1 trimers tested, but no binding was detected when engrafted on VH1-69*01?

6. There are 20 VH1-69 alleles in the IMGT database, not 17. Is the gene assignment for all the VH1-69 bnAbs correct? The authors should provide the latest information (lines 257-258 and Supplementary table 1).

7. For VH1-69 alleles, L54 is not always paired with R50 (eg. VH1-69*10 and *16). F54 and G50 are also found in VH1-69*07, *15, *18 and *20. The authors claim that the G50 but not the F54 polymorphism in VH1-69 is critical for binding of germline AR3C with a bent CDRH3, but not for HEPC74 with a straight CDRH3. It is unclear if the signatures reported are really applied to other bnAbs with a bent or a straight CDRH3.

REVIEWER COMMENTS

Reviewer #1 (Remarks to the Author):

In this manuscript, the authors investigate the suitability of naive B cells expressing germline AR3C-class B cell receptors as vaccine targets for a germline-targeting approach. They show that AR3C-class germline antibodies are able to bind (and in some cases neutralize) some E2E1 trimers (candidate immunogens). They further show that a B cell line bearing germline AR3C-class B cell receptors can become activated by some E2E1 trimers (as measured by calcium influx). Lastly, naive B cells with AR3C-like B cell receptors (VH1-69, long CDRH3, +/- CxGGxC motif) are found in healthy human subjects. This work lays useful groundwork for HCV vaccine development.

The authors also undertake a detailed characterization of the features of AR3C-class mature and germline antibodies and how those features affect antigen binding. This includes the influence of the light chain and VH1-69 restriction on E1E2 binding as well as analysis of allelic variation in VH1-69 sequence. Structural data/modeling is used to suggest mechanistic explanations for their findings (e.g., bent vs straight CDRH3 in complex with E1E2). These data are provocative but ultimately, since the work only includes 2 antibodies (and their germline equivalents), the authors' current conclusions are not supported.

We thank the reviewer for the positive overall feedback. Importantly, we have now performed experiments that provide further support for our initial conclusions (see responses to major points 1 and 2).

Major issues:

1. *The authors' claims about AR3C-class antibodies are overstated. For their analyses, they tested germline and mature variants of AR3C and HEPC74 only and it is unclear to what extent these findings are generalizable to all AR3C-class antibodies. This is especially true given that the results for AR3C and HEPC74 often differ from one another (e.g., VH1-69 dependence/allelic variation and light chain dependence to a lesser degree). Furthermore, these 2 antibodies are taken as representatives of "bent" and "straight"-type AR3C-class antibodies but again, with only 1 antibody per group, no generalizable conclusions can be drawn. The authors would ideally expand their analysis to include more AR3C-class antibodies, but if this is not tenable then, at a minimum, conclusions for this portion of the manuscript should be significantly more limited.*

This point is well taken. Germline versions of most the other AR3C-class antibodies did not react (detectably) with E2E1 trimers, complicating their analysis. Therefore, we focused our studies on AR3C and HEPC74, because the germline versions of these antibodies showed the broadest binding profile, i.e. reactivity to more than one E2E1 trimer (Figure 1a. Supplementary figure 1c). We have now extended the analysis to germline AR3A and mature AR3C and HEPC74 (new figure 6b and new supplementary figure 5c). Germline AR3A reacted only to UKNP5.2.1 E2E1 trimer (see previous Supplementary figure 1c). As with AR3C, introducing G50R (but not F54L) was sufficient to abrogate binding completely (new figure 6b). Furthermore, mature AR3C and HEPC74 reacted similarly as their germline counterparts to the G50R and F54L mutations (new figure 6b and 6c). We have added these new results to the text in the Results section (lines 308, 310-312) and Discussion (lines 388-390). We

have also slightly toned down the text in the Results (lines 286-288 and lines 310-315) and Discussion (lines 386-392). Nevertheless, the new data now allow for more generalized conclusions to be drawn compared to our initial datasets.

2. *When characterizing how various antibody features affect function, binding, but not neutralization, is tested. Ultimately, the goal is that individuals will produce AR3C-like mature antibodies that are broadly neutralizing. Therefore, it seems an oversight not to show the affect that different VH or light chains have on neutralization.*

Again the reviewer brings up a valid point. We have now performed neutralization assays with the heavy chains of (germline) AR3C and HEPC74 combined with light chain of VRC01 against a panel of seven viruses (Figure 4). Notably, neutralization was largely abrogated when the HC of HEPC74 or gl-HEPC74 was combined with the VRC01 LC (Figure 4b), while combining the VRC01 LC with (gl-)AR3C did not affect neutralization (Figure 4a). These results are in line with the binding data shown previously (also in Figure 4a and b). We have now included this data in new Figure 4a and b and added a section in the Results (line 235-243).

Minor issues

1. *The sequence data in fig 5a-b is very difficult to read. Similarly the VH data in fig 6b-c is hard to parse. Please increase the size of these figures.*

We have now rearranged figures 5 and 6, because we have added additional data to Figure 5 (as response to major comment 1). Original Figure 5 is now split into Figure 5 and 6. Figure 6 is now new Figure 7 and rearranged to improve visualization.

Reviewer #2 (Remarks to the Author):

This is a follow up to a recently published manuscript in Nature Comm by Slieden et al where they described novel immunogenic permuted E1/E2 immunogenic HCV nanoparticles that induced cross neutralizing antibodies upon immunization in rabbits. Here, they utilize the same constructs to examine their capacity to bind germline precursors of known HCV broadly neutralizing antibodies (bNAbs). The authors focus on the AR3C inferred VH1-69 germline precursors of AR3C-class bNAbs as these were the ones that showed the highest binding. They further characterize this binding and interactions. They show that these constructs can bind and activate B cells expressing inferred germline AR3C-class bNAb precursors as B cell receptors but only at the highest concentrations. They show that VH1-69 allelic variation affects germline AR3C binding. They also examine a dataset containing the human baseline antibody repertoire of eight healthy individuals and show that potential AR3C-class precursors are present at high frequency in the human B cell repertoire.

COMMENTS

The manuscript provides a novel and interesting follow-up on their previous paper and provides compelling evidence that these constructs bind and activate inferred germline precursors of the AR3C

broadly neutralizing antibodies. The interactions are well characterized which are key for further development of these constructs for vaccination. I have the following comments:

We thank the reviewer for the positive feedback.

1. *The constructs only activate germline precursor expressing B cells at the highest concentration. It is also clear from Supplementary Figure 3 that the expression of the gl precursors of HEPC74 is very low. Please discuss on how this affects the overall activation of B cells.*

The flow cytometry plots (now Supplemental Figure 3c) show that the binding of the H77- and UKNP4.1.1 probes is weaker to B cell with germline HEPC74 than for germline AR3C, probably because of lower or no affinity of (gl-)HEPC74 to these probes. The level (gl-)AR3C and (gl-)HEPC74 BCR expression during the activation assay is actually similar, because we always gate B cells for the same level of GFP, and by inference BCR expression. We have now added Supplemental Figure 3b with the GFP gating and added a line in the Results and Methods to clarify the B cell activation method (lines 556-558).

2. *Figure 5C examines all the antibodies produced but we cannot distinguish which is which. Please represent as different colours. Also, the legend mentions top and bottom panels which should be right or left.*

We have edited figure 5c (new figure 6a) and changed the figure legend text.

3. *There is yet no proof regarding the ability of these constructs to stimulate germline precursors in vivo in a suitable model that would be relevant to humans. This could be challenging, please discuss.*

We have shown that E2E1 based on circulating viral strains bind certain germline precursors and that E2E1 on nanoparticles activates B cell lines expressing the cognate BCR. These are important first steps of what will be an iterative process towards a germline-targeting HCV vaccine. First, we should improve these immunogens to broaden the number of germline V_H1-69 precursor they bind and their binding affinity for them. This will increase the likelihood for these immunogens to encounter the desired germline V_H1-69 bNAb B cells in vivo. UKNP4.1.1 E2E1 is an excellent starting point for this design challenge. Second, to better mimic the human situation, naïve primary B cells isolated from healthy individuals should be probed for binding to these immunogens (e.g. improved UKNP4.1.1 E2E1) to determine if the B cells are enriched for the desired germline V_H1-69 genes. This would require additional in-depth studies that go beyond the scope of the current study. The predictive value of such analyses was recently highlighted in the HIV-1 field: a designed germline-targeting immunogen was first used as a probe to isolate V_H1*02 germline B cells from PBMCs of healthy individuals (Jardine et al. Science 2016) and these results prompted its use in a phase I clinical trial. Importantly, this immunogen was efficient in activating the desired germline V_H1-02 B cells in 97% of immunized individuals (Leggat et al. Science 2022). We have now added a short line and references to the papers mentioned above in the discussion (lines 382-385).

MINOR COMMENTS:

There are multiple typos in the document. Please revise carefully.

We have now corrected several typos in the manuscript.

Reviewer #3 (Remarks to the Author):

“Germline-targeting” is an interesting and timely strategy for vaccine design. In this manuscript, the authors generate a panel of VH1-69-derived, AR3C-class bnAb precursors to investigate HCV immunogens capable of binding the precursors and the signatures in the precursors for binding. They detect cross-binding of their E2E1 glycoprotein trimers to AR3C and HEPC74 precursors and activation of B cells expressing these precursors by a high concentration of selected E2E1 nanoparticles, but not trimers. The authors also assessed the genetic and amino acid sequence features that affect precursor binding which include an extended CDRH3 and VH1-69 allelic polymorphism. At the end of the abstract, the authors stated the identification of critical signatures in two subclasses of the AR3C-class bNAbs that will allow refined protein design to provide a framework for germline-targeting vaccine design.

A number of areas require clarification:

- 1. The bNAbs and their precursors studied have many features and their major molecular and structural features have already been reported previously. What specific signatures reported in this study are really important in germline-targeting vaccine design, and how to implement it? The authors showed their to-be-published nanoparticles can bind and stimulate the precursors, but this should apply to any nanoparticles that present E2 multivalently. How do the results impact germline-targeting and vaccine design better than what is already known in the field?*

E2 represents only a single subunit and our E2E1 is a more complete representation of the HCV glycoprotein, although not (yet) native-like (recently published: Sliepen et al. Nat Commun. 2022). Therefore, E2E1 is probably a more suitable platform for generating a (germline-targeting) vaccine. Additionally, we identified UKNP4.1.1 as a strain that engages several inferred germline precursors of V_H1-69 bNAbs and thus holds promise as a starting point for designing a germline-targeting vaccine. Furthermore, the two-component nature of the I53-50 nanoparticles allow for novel strategies, such as mosaic nanoparticles, which is useful for steering responses towards the desired epitopes (Boyoglu-Barnum et al Nature 2021; Sliepen et al. Nat. Commun. 2022).

Structures of AR3C-class bNAbs in complex with E2 and E1E2 (Flyak et al. 2018; Kong et al. Science 2013; Tzarum et al. Sci. Adv. 2019; Torrents et al. Science 2022) are instrumental for vaccine design. However, our AR3A, AR3C and HEPC74 mutants show that these E2-Ab structures cannot always predict binding. From this structural info one would expect that F54L on the L allele is responsible for loss of AR3C binding, since L54 is located in the paratope. Instead, we found that G50R is responsible for loss of gl-AR3A and gl-AR3C binding. In contrast, gl-HEPC74 was unaffected by G50R or F54L. Since ~10-15% of the human population is homozygous for the V_H1-69 L allele these results highlight that a germline-targeting vaccine needs to engage (a pool of) AR3C-class precursors that include precursors compatible with L and F alleles, rather than focusing on a single bnAb precursor.

- 2. The study only analyzed signatures in the AR3C and HEPC74 precursors. The data show that the two antibodies are quite different in sequence (CDRH1, 2 and 3) and binding requirements (heavy chain dependence, VH1-69 background and allelic polymorphism) (Figures 3-5). There seems to be no common signatures in the two antibody precursors and how these two antibodies represent a class of VH1-69 bNAbs with a bent or a straight CDRH3, respectively.*

Therefore in their conclusion, the features that are critical for binding may not be applied to other VH1-69 bnAbs

We only studied binding properties of AR3C and HEPC74 precursors because these showed binding to more than one E2E1 immunogen. We have now also tested 50R and 54L in the context of mature AR3C and HEPC74 and the results confirmed the binding data obtained from the germline precursor versions: 50R in mature AR3C abrogated binding, while 54L did not and HEPC74 binding was not affected (new supplementary figure 5c).

UKNP5.2.1 was the only E2E1 that engaged gl-AR3A (a bent CDRH3 class). In response to this reviewer's comment, we have made F54L and G50R mutations in gl-AR3A and tested binding to UKNP5.2.1 (and UKNP2.2.1 as a negative control) and the results confirmed our findings with AR3C: G50R completely abrogated binding, while F54L did not affect binding (updated Figure 6b) and alphafold predicted a clash of gl-AR3A G50R with E2 (updated Figure 6c).

These data strengthen the notion that G50R in the L allele has a major effect on binding of bent AR3C-class bnAbs.

3. *Any explanation why some germline bnAb precursors displayed neutralizing activity in the absence of detectable binding to the soluble trimer (Supplementary Figures 1 & 2). Any detectable difference between the env glycoprotein trimer and env on HCVpp?*

Binding and neutralization of the precursor bnAbs mostly correlated. However, for gl-AR3A this correlation was a lot weaker. This might be due to differences in the epitope or glycosylation on E2E1 versus E1E2 on the HCVpp. We are working on generating designs that are a better mimic of viral E1E2 and show stronger binding to germline bnAbs. We have now added a sentence in the results that highlights this (lines 148-150) and discuss potential paths forward in the Discussion (lines 384-388 and update lines 410-416).

4. *Any explanation why the germline AR3C engrafted on VH1-2*02 and VH3-23*02 (which only has 79% and 57% sequence identity with VH1-69*01, respectively) retained binding capacity to all E2E1 trimers tested (except AMS2b), but no binding when engrafted on VH1-69 L alleles (>95% sequence identity) Figures 5a and 5b)?*

As shown on figure 6b, and further supported by supplementary figure 5c, the single G50R mutation on the *V_H1-69* alleles is enough to knock out binding of both mature and germline bnAbs with a bent CDRH3. This mutation seems to be particular to the *V_H1-69* environment. We hypothesize that the bent CDRH3 offers somehow more flexibility to the overall antibody and allows for this subclass of bnAbs to still be able to engage with the trimers. The other conformation of straight CDRH3, contrary to the bent, would be more rigid and be susceptible to big environment changes, but not to single mutations like the G50R. We have now performed additional alphafold predictions confirming these hypotheses (additional panels in Figure 6c).

5. *In the heat map summarizing ELISA binding results in Figure 5, what are the WT for gl-AR3C and gl-HEPC74, respectively? It seems AR3C is derived from VH1-69*06 and HEPC74 from VH1-69*01, respectively. However in this figure, binding of AMS2b to gl-AR3C WT and to *06 is quite different. Similarly, gl-HEPC74 WT show moderate binding to all the E2E1 trimers tested, but no binding was detected when engrafted on VH1-69*01?*

gl-AR3C and gl-HEPC74 are described in the sequence alignment on Fig. 5a and 5b. As described in literature (Tzarum et al. Sci Adv 2019), gl-AR3C is derived from $V_H1-69*06$, but contains one insertion and one substitution (ins5E and S83R). AMS2b only weakly binds gl-AR3C, which might explain why a small difference might have a relatively strong effect in ELISA. For gl-HEPC74, the WT and $V_H1-69*01$ are exactly the same and therefore we initially marked the *01 row in gray. To avoid further confusion, we have now updated the heatmap and more clearly indicated where alleles are the same as WT.

6. *There are 20 VH1-69 alleles in the IMGT database, not 17. Is the gene assignment for all the VH1-69 bnAbs correct? The authors should provide the latest information (lines 257-258 and Supplementary table 1).*

We thank the reviewer for pointing this out. We have now updated the text (line 266) and we have checked the gene assignments again and confirmed they are correct.

7. *For VH1-69 alleles, L54 is not always paired with R50 (eg. VH1-69*10 and *16). F54 and G50 are also found in VH1-69*07, *15, *18 and *20. The authors claim that the G50 but not the F54 polymorphism in VH1-69 is critical for binding of germline AR3C with a bent CDRH3, but not for HEPC74 with a straight CDRH3. It is unclear if the signatures reported are really applied to other bnAbs with a bent or a straight CDRH3.*

Indeed, G50 and F54 are not always interlinked: F54 and R50 (not G50) are found in *07, *15, *18 and *20. To isolate these allelic differences, we generated G50R, F54L and G50R+F54L mutations in the WT background. As indicated in response to comment #2 to this reviewer, we have now also made G50R, F54L and double mutations in gl-AR3A (and mature AR3C and HEPC74) and basically found the same result: G50R abrogated binding of bent CDRH3 AR3C-class bNAbs (i.e. AR3A and AR3C), but not a straight CDRH3 AR3C-class bNAbs (i.e. HEPC74).

REVIEWERS' COMMENTS

Reviewer #1 (Remarks to the Author):

Regarding the authors' response to Major Point #1:

I appreciate the inclusion of AR3A in experiments to define key characteristics of AR3C-like NAbS and understand why experiments couldn't be done with the other germline antibodies that had poor/no binding. However, analysis of 2 "bent" antibodies and 1 "straight" antibody cannot be used to draw definitive conclusions on the class in general. The conjecture that angle of CDRH3 approach may explain differences observed in AR3C-like NAbS is an intriguing explanatory model, but comparing 2 antibodies in one group to only 1 antibody in the other group is not sufficient to prove the hypothesis. It's possible, for instance, that HEPC74 is an outlier among all of the AR3C-like NAbS for reasons unrelated to CDRH3 approach. Specific claims that I believe are being mis/overstated are as follows:

1. "Furthermore, we established that the binding of inferred germline bNAbS relies heavily on the CDRH3, as shown by retention of binding after CDRH3 engraftment into different VH backgrounds" (lines 91-92): This is true for gl-AR3C but not for gl-HEPC74.
2. "Complementary targeting approaches towards VH1-69-derived precursors with both straight and bent CDRH3 would overcome the challenge of the F and L alleles" (lines 329-331): Targeting multiple rationally-selected precursors (e.g., gl-AR3C and gl-HEPC74) may overcome the challenge of the F and L alleles, but these data do not prove the functional significance of bent vs straight CDRH3 approach.
3. "Exploiting these E1E2 trimers, we characterized AR3C-class HCV bNAbS and their germline precursors and determined sequence features that are critical for binding" (lines 375-376): The features critical for binding often differ between gl-AR3C/gl-AR3A and HEPC74 so are not definitively determined for the AR3C class. It is reasonable to hypothesize that these differences may be due to the angle of CDRH3 approach, but the language should acknowledge that there is insufficient data to say this definitively.
4. "Our data confirm that AR3C-class antibodies and their precursors engage E1E2 utilizing mostly their heavy chain and showed that the CDRH3 is crucial but not exclusive" (lines 392-393): Again, the data for dependence on heavy chain and VH backbone differ between gl-AR3C and gl-HEPC74. As above, this should be acknowledged and if CDRH3 approach is invoked as the explanation, it should be done with acknowledgment that this remains a hypothesis.

Regarding the authors' response to Major Point #2:

I appreciate the inclusion of neutralization data and feel this point has been adequately addressed by the authors.

Minor comment:

Lines 253-254 state that, "VH3-23 is one of the most commonly found VH genes in the naïve human B cell repertoire (Fig. 6b)." I believe this should now be Fig. 7b after rearrangement of the figures.

Reviewer #2 (Remarks to the Author):

All my comments were properly addressed

REVIEWERS' COMMENTS

Reviewer #1 (Remarks to the Author):

Regarding the authors' response to Major Point #1:

I appreciate the inclusion of AR3A in experiments to define key characteristics of AR3C-like NAbS and understand why experiments couldn't be done with the other germline antibodies that had poor/no binding. However, analysis of 2 "bent" antibodies and 1 "straight" antibody cannot be used to draw definitive conclusions on the class in general. The conjecture that angle of CDRH3 approach may explain differences observed in AR3C-like NAbS is an intriguing explanatory model, but comparing 2 antibodies in one group to only 1 antibody in the other group is not sufficient to prove the hypothesis. It's possible, for instance, that HEPC74 is an outlier among all of the AR3C-like NAbS for reasons unrelated to CDRH3 approach. Specific claims that I believe are being mis/overstated are as follows:

We thank the reviewer for the positive overall feedback. We understand the point of view given and we have made the adjustments according to the suggestions. We hope to satisfy the reviewer with our changes (see responses).

1. *"Furthermore, we established that the binding of inferred germline bNAbS relies heavily on the CDRH3, as shown by retention of binding after CDRH3 engraftment into different VH backgrounds" (lines 91-92): This is true for gl-AR3C but not for gl-HEPC74.*

This point is well taken. We have now changed the statement to clearly indicate that this event happens in the case of gl-AR3C (line 89). We acknowledge that the statement may have been misleading previously and it was not our intention.

2. *"Complementary targeting approaches towards VH1-69-derived precursors with both straight and bent CDRH3 would overcome the challenge of the F and L alleles" (lines 329-331): Targeting multiple rationally-selected precursors (e.g., gl-AR3C and gl-HEPC74) may overcome the challenge of the F and L alleles, but these data do not prove the functional significance of bent vs straight CDRH3 approach.*

We understand this point brought up and have made adjustment in the text to satisfy the reviewer and hopefully clarify the statement. The experiments performed exchanging the VH background of either gl-AR3C or gl-HEPC74 and the understanding of the implications of the G50R mutations suggest that a germline-targeting vaccine can only succeed at overcoming the obstacles of these polymorphisms if we try to elicit a larger group of VH1-69 genes. We prove some the polymorphisms at position 50 can be potentially overcome by certain antibodies and therefore we should aim at targeting as broad as possible a pool of antibodies rather than focus on a single type.

3. *"Exploiting these E1E2 trimers, we characterized AR3C-class HCV bNAbs and their germline precursors and determined sequence features that are critical for binding" (lines 375-376): The features critical for binding often differ between gl-AR3C/gl-AR3A and HEPC74 so are not definitively determined for the AR3C class. It is reasonable to hypothesize that these differences may be due to the angle of CDRH3 approach, but the language should acknowledge that there is insufficient data to say this definitively.*

As in point number 2, we have made adjustments in the discussion section to draw less definitive conclusions. In line 372-374 we have specified that the findings regarding binding are specific to each AR3C-subclass antibodies rather than pointing towards a generalized mechanism. It is important to note that the hypothesis that the CDRH3 subclasses (directly linked to angle of approach) has been used in different studies and we tried in this study to extend the knowledge on these subclasses. We think the language over the length of the manuscript is in line with previous studies and we hope to have made it clear that is a hypothesis.

4. *"Our data confirm that AR3C-class antibodies and their precursors engage E1E2 utilizing mostly their heavy chain and showed that the CDRH3 is crucial but not exclusive" (lines 392-393): Again, the data for dependence on heavy chain and VH backbone differ between gl-AR3C and gl-HEPC74. As above, this should be acknowledged and if CDRH3 approach is invoked as the explanation, it should be done with acknowledgment that this remains a hypothesis.*

In line with the previous points, we have adjusted the text in lines 391-392 to clarify that there are differences between gl-AR3C and gl-HEPC74 and to not give overall conclusions regarding the AR3C-class bNAbs.

- *Regarding the authors' response to Major Point #2: I appreciate the inclusion of neutralization data and feel this point has been adequately addressed by the authors.*

We thank the reviewer for the feedback on major point #2 and happy to satisfy comments addressed previously regarding the inclusion of further supporting data.

Minor comment:

- *Lines 253-254 state that, "VH3-23 is one of the most commonly found VH genes in the naïve human B cell repertoire (Fig. 6b)." I believe this should now be Fig. 7b after rearrangement of the figures.*

After rearrangement of the figures we did not realize this mistake. We have changed it so that the text corresponds to the referred figure (line 251).

Reviewer #2 (Remarks to the Author):

All my comments were properly addressed

We thank the reviewer for the positive feedback and we are happy to have successfully addressed his or her concerns.